# State-space kinetic Ising model reveals task-dependent entropy flow in sparsely active nonequilibrium neuronal dynamics

**Ken Ishihara**[1,2] ✉ **& Hideaki Shimazaki** [2,3] ✉

Neuronal ensemble activity, including coordinated and oscillatory patterns, exhibits hallmarks of nonequilibrium systems with time-asymmetric trajectories to maintain their organization. However, assessing time asymmetry from neuronal spiking activity remains challenging. The kinetic Ising model provides a framework for studying the causal, nonequilibrium dynamics in spiking recurrent neural networks. Recent theoretical advances in this model have enabled time-asymmetry estimation from large-scale steady-state data. Yet, neuronal activity often exhibits time-varying firing rates and coupling strengths, violating the steady-state assumption. To overcome this limitation, we developed a state-space kinetic Ising model that accounts for nonstationary and nonequilibrium properties of neural systems. This approach incorporates a mean-field method for estimating time-varying entropy flow, a key measure for maintaining the system's organization through dissipation. Applying this method to mouse visual cortex data revealed greater variability in causal couplings during task engagement despite reduced neuronal activity with increased sparsity. Moreover, higher-performing mice exhibited increased coupling-related entropy flow per spike during task engagement, suggesting more efficient computation in the higher-performing mice. These findings underscore the model's utility in uncovering intricate asymmetric causal dynamics in neuronal ensembles and linking them to behavior through the thermodynamic underpinnings of neural computation.

The emergence of ordered spatiotemporal dynamics in nonequilibrium systems that continuously exchange energy and matter with their surroundings has intrigued many scientists[1–5], as it provides a foundational mechanism for phenomena such as chemical oscillations, morphogenesis, and collective behaviors like animal herding. Nonequilibrium processes inherently violate the detailed balance between the forward and reverse transitions, yielding time-asymmetric, irreversible dynamics. Stochastic thermodynamics has clarified that this time-asymmetry is essential for systems to sustain their organized structure by dissipating entropy into the environment[6–9].

Further, the thermodynamic uncertainty relation[10–13] and the thermodynamic speed-limit theorem[14,15] show that dissipation sets fundamental bounds on how precisely and rapidly systems can evolve.

Neural systems are no exception. In animals engaged in behavioral and cognitive tasks, the dynamics of neuronal population activity exhibit hallmarks of nonequilibrium systems. Notable examples include the rotational activity of M1 neurons during motor execution tasks[16,17] and the sequential patterns observed in hippocampal neurons, including their replay, during navigation and sleep[18–20]. Since the original proposal of cell assembly and its phase

[1]Graduate School of Life Science, Hokkaido University, Sapporo, Japan. [2]Center for Human Nature, Artificial Intelligence, and Neuroscience (CHAIN), Hokkaido University, Sapporo, Japan. [3]Graduate School of Informatics, Kyoto University, Kyoto, Japan. ✉e-mail: ishihara.ken.n7@elms.hokudai.ac.jp; h.shimazaki@i.kyoto-u.ac.jp

sequences by Donald O. Hebb[21], coordinated sequential patterns have been thought fundamental for memory consolidation and retrieval[22–25]. Recently, studies on fMRI or ECoG suggested that increased time-asymmetry in neural signals, quantified by steady-state entropy production[7,26,27], could serve as a signature of consciousness[28–31] or reflect the cognitive load demanded by tasks[32]. For instance, entropy production measured from ECoG signals of non-human primates is diminished during sleep and certain types of anesthesia compared to wakefulness[28,31], indicating that the awake state includes more directed temporal patterns. However, assessing entropy production directly from neuronal spiking activities remains challenging. Further complicating this issue, neural signals exhibit nonstationary dynamics, including transient or oscillatory behavior, which hinders the use of steady-state entropy production metrics.

The kinetic Ising model is a prototypical model of recurrent neural networks[33,34]. It extends the equilibrium Ising model, which has been successfully applied to empirical spiking data to elucidate the thermo-dynamic and associative-memory properties of neural systems[35,36]. In the kinetic Ising system, neurons are causally driven by the past states of self and other neurons, as well as a force representing the intrinsic excitability of the neurons and/or an influence of unobserved concurrent signals. When neurons receive steady inputs and their causal couplings are asymmetric, the system does not relax to an equilibrium state. Instead, it exhibits steady-state nonequilibrium dynamics characterized by non-zero entropy production. Recent theoretical studies on steady-state entropy production have elucidated its behavior in relation to distinct phases of the Ising system, including critical phase transitions[37]. Mean-field theories have been developed for kinetic Ising systems[38–43], enabling the estimation of steady-state entropy production from large-scale spike sequences[43]. However, neuronal activity exhibits dynamical changes not only in firing rates but also in the strength of their interactions, both of which violate the steady-state assumptions.

To account for the nonstationary dynamics of neural systems, the state-space method[44,45] has been applied to the Ising system[46–50]. In these approaches, Bayesian filtering and smoothing algorithms have been developed to estimate time-dependent parameters of the Ising model, along with an EM algorithm[51,52] to optimize various hyper-parameters. These models have enabled researchers to trace time-varying neuronal interactions while neurons' internal parameters change dynamically, absorbing the effect from unobserved concurrent signals. Additionally, it has elucidated the thermodynamic quantities of neural systems (e.g., free energy and specific heat) in a time-dependent manner, in relation to the behavioral paradigms of tasks[48]. Nevertheless, these methods assume an equilibrium Ising model with symmetric couplings, which limits their ability to assess the none-quilibrium properties of observed neural activities.

In this study, we develop the state-space kinetic Ising model to account for the nonstationary and nonequilibrium properties of neural activities. We also construct a mean-field method for estimating time-varying entropy flow, an essential component of entropy production that quantifies the dissipation of entropy, from spiking activities of neural ensembles. Given this method, we hypothesize that entropy flow, as estimated with the kinetic Ising framework, reflects the capacity of neural populations to perform meaningful computation under energetic and behavioral-time constraints[53]. Specifically, we expect that high-performing animals would exhibit greater entropy flow per spike, consistent with efficient coding.

Application of the methods to mouse V1 neurons revealed behavior-dependent changes in entropy flow. From the analysis of 37 mice, we found that while spike rates of the populations are lower on average and exhibited sparser distributions when mice actively engaged in tasks than in the passive condition, active engagement significantly enhanced the variability of the neuronal couplings, which contributed to increasing entropy flow. Further, higher-performing mice exhibited stronger entropy flow per spike in active engagement

than in the passive condition. We corroborated contributions of couplings to this tendency using trial-shuffled data that excluded influences of firing rate dynamics and sampling errors in estimating neuronal couplings. Thus, the method enabled us to reveal contributions of behavior-related neuronal couplings to the causal activities in sparsely active neuronal populations, while isolating firing rate dynamics. These results imply economical representations of stimuli by time-asymmetric causal activity in competent mice.

This paper is organized as follows. In Results, we first introduce the state-space kinetic Ising model and its estimation method. Next, we introduce the mean-field method for estimating entropy flow. We validate these methods through simulations and then apply them to mouse V1 data. Finally, we relate our findings to previous studies and discuss their implications for efficient information coding in neural populations.

## Results

### The state-space kinetic Ising model

In neurophysiological experiments, the experimentalists simultaneously record the activity of multiple neurons while animals are exposed to a stimulus or perform a task, and repeat the recordings multiple times under the same experimental conditions. We analyze the quasi-simultaneous activity of neurons using binarized spike sequences. For this goal, we convert the simultaneous sequences of spike timings of $N$ neurons into sequences of binary patterns by binning them with a bin width of $\Delta$ [ms]. We assign a value of 1 if there is one or more spikes in a bin and 0 otherwise. We assume that there are $T + 1$ bins for each trial, with an initial bin being the 0th bin and $L$ trials in total. Below, we treat the bins as discrete time steps and refer to the $t$th bin as time $t$. We let $x_{i,t}^l = \{0, 1\}$ be a binary variable of the $i$th neuron at time $t$ in the $l$th trial ($i = 1,...,N$, $t = 0,..., T$, $l = 1,...,L$). We collectively denote the binary patterns of simultaneously recorded neurons at time $t$ in the $l$th trial using a vector, $\mathbf{x}_t^l = (x_{1,t}^l, \ldots, x_{i,t}^l, \ldots, x_{N,t}^l)$. Further, we denote the patterns at time $t$ from all trials by $\mathbf{x}_t = (\mathbf{x}_t^1, \ldots, \mathbf{x}_t^l, \ldots, \mathbf{x}_t^L)$ and denote all the patterns up to time $t$ by $\mathbf{x}_{0:t}$.

We construct the state-space kinetic Ising model to account for the nonequilibrium dynamics of the binary sequences by extending the state-space models developed for equilibrium Ising systems[47,48]. The state-space model is composed of the observation model and the state model. The observation model in the $t$th bin is

$$p(\mathbf{x}_t | \mathbf{x}_{t-1}, \boldsymbol{\theta}_t) = \prod_{l=1}^{L} \prod_{i=1}^{N} p(x_{i,t}^l | \mathbf{x}_{t-1}^l, \boldsymbol{\theta}_t^i)$$
$$= \prod_{l=1}^{L} \prod_{i=1}^{N} \exp\left[ \theta_{i,t} x_{i,t}^l + \sum_{j=1}^{N} \theta_{ij,t} x_{i,t}^l x_{j,t-1}^l - \psi(\boldsymbol{\theta}_t^i, \mathbf{x}_{t-1}^l) \right], \quad (1)$$

where $\theta_{i,t}$ is a time-dependent (external) field parameter that determines the bias for inputs to the $i$th neuron at time $t$ and $\theta_{ij,t}$ is a time-dependent coupling parameter from the $j$th neuron to the $i$th neuron. These parameters are collectively denoted as $\boldsymbol{\theta}_t = (\boldsymbol{\theta}_t^1, \ldots, \boldsymbol{\theta}_t^i, \ldots, \boldsymbol{\theta}_t^N)$ and $\boldsymbol{\theta}_t^i = (\theta_{i,t}, \theta_{i1,t}, \ldots \theta_{ij,t}, \ldots, \theta_{iN,t})$. $\psi(\boldsymbol{\theta}_t^i, \mathbf{x}_{t-1}^l)$ is the log normalization term defined as

$$\psi(\boldsymbol{\theta}_t^i, \mathbf{x}_{t-1}^l) = \log\left[ 1 + \exp\left[ \theta_{i,t} + \sum_{j=1}^{N} \theta_{ij,t} x_{j,t-1}^l \right] \right]. \quad (2)$$

We also specify $p(\mathbf{x}_0)$, a probability mass function of the binary patterns at $t = 0$, which we assume $p(\mathbf{x}_0) = \prod_{i=1}^{N} \prod_{l=1}^{L} p(x_{i,0}^l)$, where $p(x_{i,0}^l) = 0.5$ for data generation.

Next, we introduce a state model of the time-varying parameters $\boldsymbol{\theta}_t$ for $t = 1,...,T$:

$$p(\boldsymbol{\theta}_{1:T} | \mathbf{w}) = \prod_{i=1}^{N} \left[ p(\boldsymbol{\theta}_1^i | \boldsymbol{\mu}^i, \Sigma^i) \prod_{t=2}^{T} p(\boldsymbol{\theta}_t^i | \boldsymbol{\theta}_{t-1}^i, \mathbf{Q}^i) \right], \quad (3)$$

where $\mathbf{w}$ denotes the collection of the hyperparameters: $\mathbf{w} = [\boldsymbol{\mu}^1,...,\boldsymbol{\mu}^N, \boldsymbol{\Sigma}^1,...,\boldsymbol{\Sigma}^N, \mathbf{Q}^1,...,\mathbf{Q}^N]$. Namely, we assume independence of the parameters of a neuron from those of the other neurons, which significantly reduces computational costs. The transition of the $i$th neuron follows the linear Gaussian models:

$$p(\boldsymbol{\theta}_t^i|\boldsymbol{\theta}_{t-1}^i, \mathbf{Q}^i) = \frac{1}{\sqrt{|2\pi\mathbf{Q}^i|}} \exp\left[\frac{1}{2}(\boldsymbol{\theta}_t^i - \boldsymbol{\theta}_{t-1}^i)^\top (\mathbf{Q}^i)^{-1}(\boldsymbol{\theta}_t^i - \boldsymbol{\theta}_{t-1}^i)\right], \quad (4)$$

while the initial density $p(\boldsymbol{\theta}_1^i|\boldsymbol{\mu}^i, \boldsymbol{\Sigma}^i)$ is given by the Gaussian distribution with mean $\boldsymbol{\mu}^i$ and covariance $\boldsymbol{\Sigma}^i$.

## Model fitting and inference
Our goal is to obtain the approximation of the posterior density of the trajectory $\boldsymbol{\theta}_{1:T}$ given the observed neural activity $\mathbf{x}_{0:T}$:

$$p(\boldsymbol{\theta}_{1:T}|\mathbf{x}_{0:T}, \mathbf{w}) = \frac{p(\mathbf{x}_{0:T}|\boldsymbol{\theta}_{1:T})p(\boldsymbol{\theta}_{1:T}|\mathbf{w})}{p(\mathbf{x}_{0:T}|\mathbf{w})}, \quad (5)$$

while optimizing the hyperparameters $\mathbf{w}$ under the principle of maximizing marginal likelihood:

$$p(\mathbf{x}_{0:T}|\mathbf{w}) = p(\mathbf{x}_0)\prod_{t=1}^{T} p(\mathbf{x}_t|\mathbf{x}_{0:t-1}, \mathbf{w})$$
$$= p(\mathbf{x}_0)\prod_{t=1}^{T}\prod_{l=1}^{L}\prod_{i=1}^{N}\int p(x_{i,t}^l|\mathbf{x}_{t-1}^l, \boldsymbol{\theta}_t^i)p(\boldsymbol{\theta}_t^i|\mathbf{x}_{0:t-1}^l, \mathbf{w})\,d\boldsymbol{\theta}_t^i. \quad (6)$$

Here, $p(\boldsymbol{\theta}_t^i|\mathbf{x}_{0:t-1}^l, \mathbf{w})$ is the one-step prediction density.

The Expectation-Maximization (EM) algorithm[54] offers a way to construct the approximate posterior with optimized hyperparameters by alternately constructing the approximate posterior density while fixing the hyperparameters (E-step) and optimizing the hyperparameters while fixing the approximate posterior (M-step). The construction of the approximate posterior density at the E-step is performed by sequentially applying Bayes algorithms in a forward and backward manner, where we approximate the posteriors by Gaussian distributions using Laplace's method. Thus, the method yields the mean and variance of the approximated Gaussian posterior at time $t$, which are denoted as $\boldsymbol{\theta}_{t|T}$ and $\mathbf{W}_{t|T}$, respectively. See Methods and Supplementary Note 1 for the details of the algorithm.

## Entropy flow
Using the inferred parameters $\boldsymbol{\theta}_{1:T}$ of the kinetic Ising model from spike data, we estimate entropy flow (also known as bath entropy change) at each time step. The entropy flow at time $t$ is defined as

$$\sigma_t^{\text{flow}} = \sum_{\mathbf{x}_t, \mathbf{x}_{t-1}} p(\mathbf{x}_t, \mathbf{x}_{t-1}) \log\frac{p(\mathbf{x}_t|\mathbf{x}_{t-1})}{p(\mathbf{x}_{t-1}|\mathbf{x}_t)}, \quad (7)$$

where $p(\mathbf{x}_{t-1}|\mathbf{x}_t)$ represents the probability of observing time-reversed processes generated under the forward model. Because we use the natural logarithm, we report entropy flow in units of nats. Eq. (7) is related to the entropy production $\sigma_t$ at time $t$ as follows[7,9,26,27]:

$$\sigma_t = \sum_{\mathbf{x}_t, \mathbf{x}_{t-1}} p(\mathbf{x}_t, \mathbf{x}_{t-1}) \log\frac{p(\mathbf{x}_t|\mathbf{x}_{t-1})p_{t-1}(\mathbf{x}_{t-1})}{p(\mathbf{x}_{t-1}|\mathbf{x}_t)p_t(\mathbf{x}_t)}$$
$$= (S_t - S_{t-1}) + \sigma_t^{\text{flow}}. \quad (8)$$

Here $p_t(\mathbf{x}_t)$ is the marginal probability mass function of the system at time $t$. $S_t$ is the entropy of the system at time $t$ defined as

$$S_t = -\sum_{\mathbf{x}_t} p_t(\mathbf{x}_t)\log p_t(\mathbf{x}_t). \quad (9)$$

The entropy production is non-negative: $\sigma_t \geq 0$. Thus, the positive entropy flow allows a decrease in the system's entropy: namely, the system can be more structured or organized when the entropy flow is positive. Since it is challenging to estimate the system's entropy or its change, here we estimate the entropy flow, which provides the lower bound of the entropy change: $S_t - S_{t-1} \geq -\sigma_t^{\text{flow}}$.

Similarly, since the total entropy production across all time steps $\sigma_{1:T}$ is given as

$$\sigma_{1:T} = \sum_{t=1}^{T}\sigma_t = (S_T - S_0) + \sum_{t=1}^{T}\sigma_t^{\text{flow}}, \quad (10)$$

the total entropy flow $\sum_{t=1}^{T}\sigma_t^{\text{flow}}$ provides the lower bound of the system's entropy change from the initial and final time step: $S_T - S_0 \geq -\sum_{t=1}^{T}\sigma_t^{\text{flow}}$. This indicates that the positive total entropy flow enables the systems to be more structured (i.e., lower entropy) at the final time step than at the initial time step.

In this study, we refer to Eq. (7) as entropy flow because it is related to heat flow to reservoirs (thermal bath) and the entropy change of the reservoirs in thermodynamics[53]. We note that Eq. (7) differs from the entropy flow defined elsewhere[55,56], which was obtained by the decomposition of the dissipation function[27] as an alternative to entropy production. See refs. 27,57 for their distinct definitions and decompositions for the case of discrete-time systems.

For the case of the kinetic Ising model, the entropy flow is written as

$$\sigma_t^{\text{flow}} = \sum_{i=1}^{N}\theta_{i,t}\left(E_{\mathbf{x}_t}x_{i,t} - E_{\mathbf{x}_{t-1}}x_{i,t-1}\right)$$
$$+ \sum_{i,j}\theta_{ij,t}E_{\mathbf{x}_t, \mathbf{x}_{t-1}}\left(x_{i,t}x_{j,t-1} - x_{i,t-1}x_{j,t}\right) \quad (11)$$
$$- \sum_{i=1}^{N}\left(E_{\mathbf{x}_{t-1}}\psi(\boldsymbol{\theta}_t^i, \mathbf{x}_{t-1}) - E_{\mathbf{x}_t}\psi(\boldsymbol{\theta}_{t-1}^i, \mathbf{x}_t)\right),$$

where $E_{\mathbf{x}_t}$ and $E_{\mathbf{x}_t, \mathbf{x}_{t-1}}$ represents expectation by $p(\mathbf{x}_t)$ and $p(\mathbf{x}_t, \mathbf{x}_{t-1})$, respectively.

## Mean-field estimation of entropy flow
Entropy flow (Eq. (7)) requires the expectation by the joint density $p(\mathbf{x}_t, \mathbf{x}_{t-1})$, which is computationally expensive for large systems. While the mean-field methods for the kinetic Ising model[38–43] were employed to estimate steady-state entropy flow[43], the mean-field method for estimating time-varying entropy flow remains unexplored. Here, we develop the mean-field method for estimating dynamic entropy flow.

The entropy flow $\sigma_t^{\text{flow}}$ can be decomposed into the forward and reverse components,

$$\sigma_t^{\text{flow}} = -\sigma_t^{\text{forward}} + \sigma_t^{\text{backward}}, \quad (12)$$

where $\sigma_t^{\text{forward}}$ and $\sigma_t^{\text{backward}}$ denote the conditional entropies of the forward and time-reversed conditional distributions, respectively. The proposed mean-field method estimates the entropy flow by approximating the forward and time-reversed conditional entropies using the Gaussian integral:

$$\sigma_t^{\text{forward}} \approx \sum_{i=1}^{N}\int \mathcal{D}z\,\chi\left(g_{i,t,t-1} + z\sqrt{\Delta_{i,t,t-1}}\right), \quad (13)$$

$$\sigma_t^{\text{backward}} \approx \sum_{i=1}^{N} \int \mathcal{D}_z\, \phi_{i,t}\left(g_{i,t,t} + z\sqrt{\Delta_{i,t,t}}\right), \qquad (14)$$

where $\mathcal{D}_z = \frac{dz}{\sqrt{2\pi}}\exp\left(-\frac{1}{2}z^2\right)$. See Methods and Supplementary Note 2 for the derivation of these results. The functions $\chi(h)$ and $\phi_{i,t}(h)$ are given as follows. $\chi(h)$ is entropy of $(0, 1)$ binary random variables with mean $r(h) = 1/(1 + e^{-h})$:

$$\chi(h) = -r(h)h + \psi(h), \qquad (15)$$

where we redefined the log normalization function $\psi$ as a function of $h$: $\psi(h) = \log(1 + e^h)$. $\phi_{i,t}(h)$ is given by

$$\phi_{i,t}(h) = -m_{i,t-1}h + \psi(h), \qquad (16)$$

where $m_{i,t-1}$ is the mean-field activation rate of $i$th neuron at time $t - 1$ (see below for how to obtain it).

Here, the input $h = g_{i,t,s} + z\sqrt{\Delta_{i,t,s}}$ is a Gaussian random variable with mean $g_{i,t,s}$ and variance $\Delta_{i,t,s}$ ($s = t, t - 1$), where $z$ denotes a standardized Gaussian random variable. $g_{i,t,s}$ and $\Delta_{i,t,s}$ are computed using the mean-field activation rate at time $s$, $m_{i,s}$, as

$$g_{i,t,s} = \theta_{i,t} + \sum_j \theta_{ij,t} m_{j,s}, \qquad (17)$$

$$\Delta_{i,t,s} = \sum_j \theta_{ij,t}^2 m_{j,s}(1 - m_{j,s}). \qquad (18)$$

The mean-field activation rate $m_{i,t}$ can be recursively computed using

$$m_{i,t} \approx \int \mathcal{D}_z\, r\left(g_{i,t,t-1} + z\sqrt{\Delta_{i,t,t-1}}\right), \qquad (19)$$

starting with nominal values of $m_{i,0}$. In this study, we use spiking probability averaged over all time bins and trials for each neuron as $m_{i,0}$.

We also note that under the steady-state assumption, the mean-field approximation can be expressed using the stationary parameters $m_i$, $g_i$, and $\Delta_i$ as (See Supplementary Note 3):

$$\sigma_t^{\text{flow}} \approx \sum_{i=1}^{N} \int \mathcal{D}_z\left(r(g_i + z\sqrt{\Delta_i}) - m_i\right) \cdot z\sqrt{\Delta_i}. \qquad (20)$$

The term $r(g_i + z\sqrt{\Delta_i}) - m_i$ quantifies how the neuron's activity rate deviates from its long-term average, while $z\sqrt{\Delta_i}$ represents the fluctuations of the input it receives. The steady-state mean-field solution thus provides an intuitive view of entropy flow as a measure of a neuron's causal responsiveness to input fluctuations—a quantity that captures the correlation underlying Hebbian plasticity in neural systems. However, this equation also clarifies that the approximation depends mainly on the magnitudes of the field and coupling parameters and is thus insensitive to the detailed coupling structure. It should therefore be applied with caution when the degree of coupling asymmetry is the primary determinant of the strength of entropy flow.

## Simulation: estimating the model parameters

We begin by testing the proposed method by estimating the time-dependent parameters of a kinetic Ising model consisting of two simulated neurons (Fig. 1A). Figure 1B shows the spike data generated using Eq. (1) with the number of bins, $T = 400$, and the number of trials, $L = 200$. The time-dependent parameters $\boldsymbol{\theta}_{1:T}$ used to generate binary data were sampled from Gaussian processes (See Methods).

The EM algorithm was applied to this spike data until the log marginal likelihood converged (Fig. 1C). Figure 1D shows the components of the optimized hyperparameter matrices, $\mathbf{Q}^i(i = 1, 2)$. Figure 1E shows the MAP estimates of the time-dependent fields $\theta_{i,t}$ and couplings $\theta_{ij,t}$ under the optimized hyperparameters (solid lines) with 95% credible intervals (shaded areas). The results confirm that the method uncovers the underlying time-dependent parameters (black dashed lines) used to generate the data.

Next, we applied the state-space kinetic Ising model to a network of 12 simulated neurons to estimate the time-varying field and coupling parameters between neurons. Figure 2A presents the spike data generated using the observation model with the number of bins set to $T = 75$ and the number of trials $L = 200$. Data generation and model estimation procedures follow the two-neuron case above. Figure 2B shows the estimated time-varying coupling parameters $\theta_{ij,t}$ for each neuron. In Fig. 2C, we compare the estimated coupling parameters $\boldsymbol{\theta}_{t|T}$ with the true values $\boldsymbol{\theta}_t$ at representative time points ($t = 10, 20,..., 60$). The scatter plot shows agreement between the true and estimated values, with most points aligning closely along the diagonal line, indicating that the model captured the underlying dynamics of the coupling parameters. These results confirm that the proposed state-space kinetic Ising model can reliably estimate time-varying coupling parameters in a network of simulated neurons.

## Simulation: estimation error and computational time

We evaluated the performance of the proposed state-space kinetic Ising model in terms of estimation accuracy and computational time, varying dataset and population sizes (See Methods for parameter generation).

Estimation error: To assess estimation error, we computed the root mean squared error (RMSE) between the true parameters $\boldsymbol{\theta}_t$ and the estimated parameters $\boldsymbol{\theta}_{t|T}$ for both field and coupling parameters. Namely, RMSEs were computed separately for the field parameter $\theta_{i,t}$ and the coupling parameter $\theta_{ij,t}$, then averaged over time bins. The means over 10 independent samplings are shown in Fig. 3A, B with the standard deviations represented by error bars.

For a fixed number of neurons ($N = 80$), RMSEs for both field and coupling parameters decreased as the number of trials $L$ increased (Fig. 3A), demonstrating improved estimation accuracy with more data. Conversely, when the number of trials was fixed at $L = 550$, RMSEs exhibited different trends depending on the parameter type. The RMSE for the field parameter increased with $N$, imposing the challenges of estimating field parameters in larger networks with limited data. The RMSE for coupling parameters remained stable across different neuron numbers in this simulation (Fig. 3B).

Computational time: We analyzed the computation time for model fitting. Figure 3C illustrates the computation time required to complete the EM algorithm for different numbers of neurons $N$ and trials $L$. The results indicate that estimation with $N = 80$ and $L = 550$ trials can be completed in approximately one hour, making it feasible for practical data analysis. Nevertheless, computation time scales with both $N$ and $L$, highlighting the necessity for further optimization to enable large-scale analysis. The assumption of independent state evolution for individual neurons (Eq. (3)) significantly reduces computational complexity by enabling independent calculations for filtering, smoothing, and parameter optimization per neuron, which can be further accelerated through parallel updates. Another potential improvement is replacing the current filtering method, which employs exact Newton-Raphson optimization for maximum a posteriori (MAP) estimation, with quasi-Newton or mean-field approximations, as demonstrated in equilibrium state-space Ising models[48].

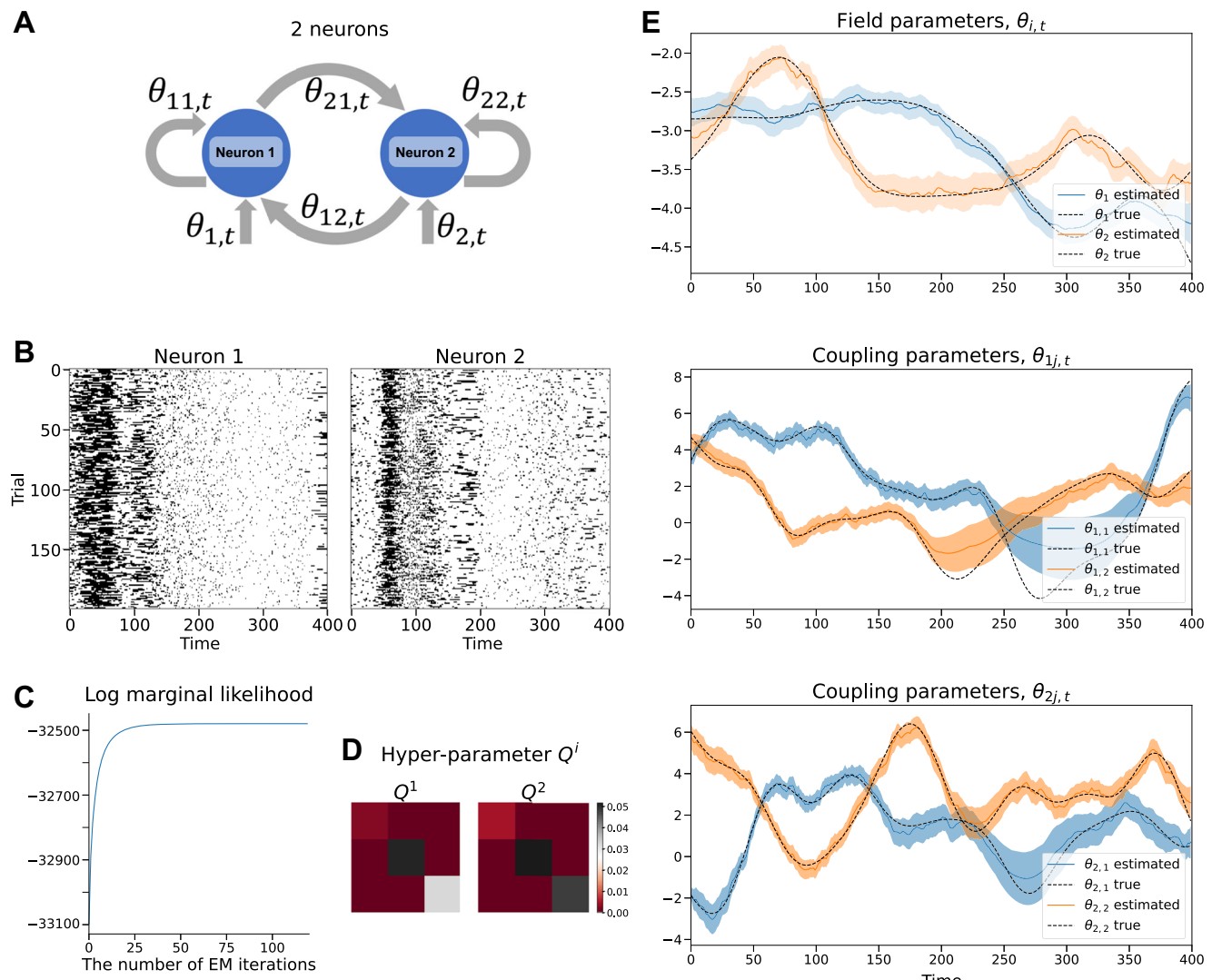

**Fig. 1 | Application of the state-space kinetic Ising model to two simulated neurons. A** A schematic of the time-dependent kinetic Ising model for two neurons with field and coupling parameters. The links between the nodes represent the neurons' causal interactions with arrows indicating the time evolution from the past to the present. **B** Raster plots for the two neurons. The vertical axis represents the number of trials, and the horizontal axis shows the number of time bins. **C** The approximate marginal log-likelihood as a function of the iteration steps of the EM

algorithm. **D** The optimized hyperparameter $\mathbf{Q}^i$ for neuron 1 (left) and neuron 2 (right). **E** (top) Estimated and true time-dependent field parameters. The solid lines represent the MAP estimates of the field (first-order) parameters obtained from the smoothing posterior, $\boldsymbol{\theta}_{t|T}$. The shaded areas show the 95% credible intervals derived from the diagonal elements of the smoothed covariance matrix, $\mathbf{W}_{t|T}$. The dotted lines are the field parameters from true $\boldsymbol{\theta}_t$ used to generate the data. (middle, bottom) Estimated and true time-dependent coupling (second-order) parameters.

## Simulation: estimating entropy flow

In this section, we assess the proposed mean-field approximation method for estimating entropy flow. As in the previous section, we generated spike samples from time-dependent parameters $\boldsymbol{\theta}_{1:T}$ sampled from Gaussian processes. All simulations were conducted with $N = 80$, $T = 75$, and $L = 550$ trials. We then estimated the time-dependent field and coupling parameters from the data. Using the posterior mean $\boldsymbol{\theta}_{t|T}$, we obtained the mean-field approximation of the time-dependent entropy flow (Eq. (12), using Eqs. (13) and (14)). The solid red line in Fig. 4 represents the entropy flow calculated using the mean-field approximation with the learned parameters.

To verify the consistency of the estimated entropy flow, we calculated the entropy flow using a sampling-based method to compute the expectation over the two-step trajectories (solid black). This approach involves repeatedly running the kinetic Ising model (Eq. (1)) using the true parameters to sample binary spike sequences. This process was performed $n_s = 10,000$ times to empirically estimate the

joint distribution $p(\mathbf{x}_t, \mathbf{x}_{t-1})$. Using this empirical distribution, we obtained a sample estimate of the entropy flow as follows:

$$\hat{\sigma}_t^{\text{flow}} = \frac{1}{n_s}\sum_{s=1}^{n_s} \log \frac{p(\mathbf{x}_t^s | \mathbf{x}_{t-1}^s)}{p(\mathbf{x}_{t-1}^s | \mathbf{x}_t^s)}, \quad (21)$$

where $\mathbf{x}_t^s$ denotes the $s$th sample at time $t$. This sampling estimation using the true parameters serves as the baseline.

The mean-field estimation of the entropy flow (solid red) follows the trajectory of the baseline sampling estimation using the true parameters (solid black). The result confirms that the proposed method is applicable for entropy flow analysis while ensuring computational feasibility. The slight discrepancy between the two lines is due to the errors in estimating the time-dependent parameters and/or the mean-field approximation (in addition to sampling fluctuation inherent to the sampling method). To separate these effects, we estimated the entropy flow by the mean-field approximation using the

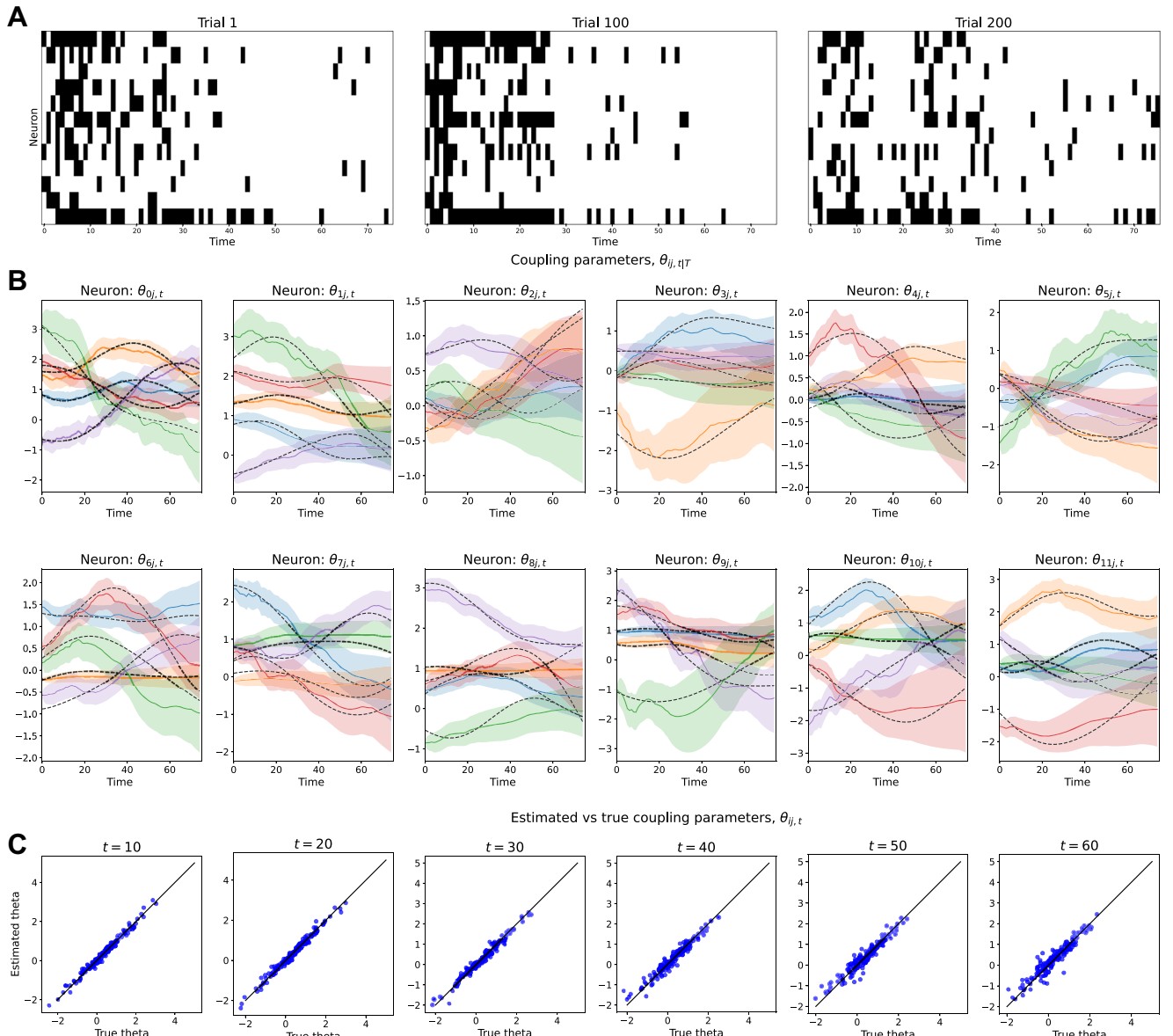

**Fig. 2 | The application of the state-space kinetic Ising model to 12 simulated neurons.** **A** Simulated spike data for the first, 100th, and last trial out of 200 trials. The vertical axis shows the number of neurons, and the horizontal axis represents the number of bins. **B** Estimated coupling parameters $\boldsymbol{\theta}_{t|T}$ (solid lines), for all neurons and time bins ($i = 1, 2,..., 12, t = 1,..., T$). Shaded areas indicate 95% credible intervals, and dashed lines denote the true parameter values used to generate the data. These plots show only the couplings that are significantly deviated from zero: The couplings whose 95% credible interval contains 0 in all bins were excluded. For clarity, only five such significant incoming couplings from other neurons are shown in each panel. **C** Scatter plots comparing the true coupling parameters $\boldsymbol{\theta}_t$ with the estimated values $\boldsymbol{\theta}_{t|T}$ at time $t = 10, 20,...,60$. The black line is a diagonal line.

true parameters $\boldsymbol{\theta}_{1:T}$ used for the data generation (dashed blue). This estimation deviated from the baseline sampling estimation. In contrast, the sampling method using estimated parameters (dashed green) did not significantly differ from the baseline. Thus, the discrepancy arose from the mean-field approximation, rather than from inaccuracies in parameter estimation. These results suggest that refining the mean-field method could further improve the accuracy of entropy flow estimation.

## Simulation: model limitations

We end the simulation analysis by acknowledging that assumptions of the kinetic Ising framework, in particular pairwise couplings and conditional independence, represent simplifications that may not faithfully capture neural population dynamics. To demonstrate this, we fitted the kinetic Ising model to population activity, using a neuronal population model called the alternating-shrinking higher-order interaction model, which accounts for deviations from the logistic activation function of individual neurons and exhibits higher-order interactions[58].

In this model, homogeneous binary population activity was generated using an exponential-family distribution with interactions of all orders (Eq. (63) in Methods). The model was designed so that the spike-count histogram of the population exhibits sparse yet widespread characteristics (Fig. 5A, green), consistent with empirical data. We performed Gibbs sampling from this distribution (blue circle), which corresponds to the dynamics of recurrent networks with an extended activation function (See Methods).

When the state-space kinetic Ising model was fitted to these activities, it failed to reproduce the observed spike-count histogram (Fig. 5A, yellow). One reason is its restriction to pairwise interactions,

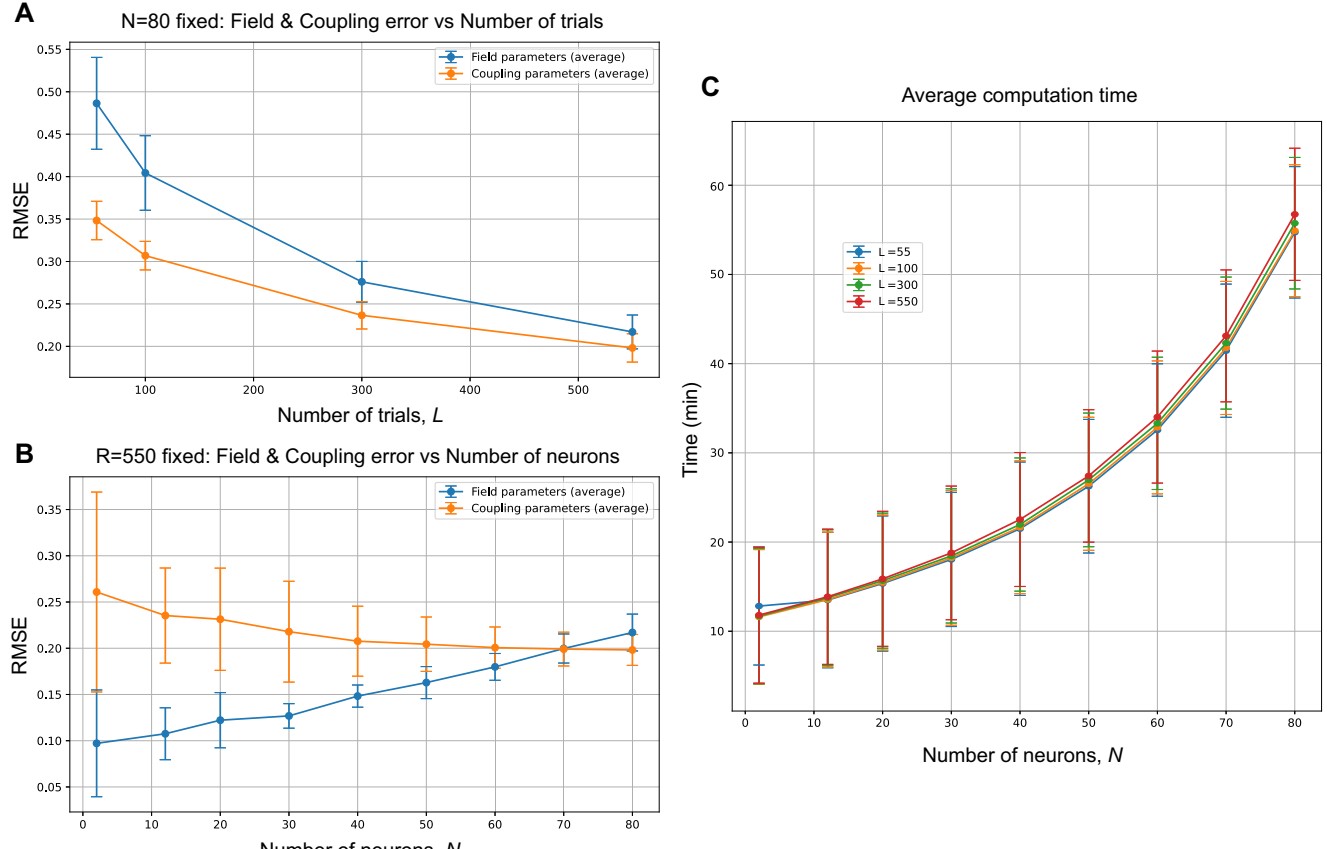

**Fig. 3 | Estimation error and computational time. A** Root mean squared error (RMSE) of the field and coupling parameter estimation as a function of trials $L$, with the number of neurons fixed at $N = 80$. Results are averaged over 10 independent samples, with error bars representing standard deviations. **B** RMSE of the field and coupling parameters as a function of the number of neurons $N$, with the number of trials fixed at $L = 550$. Averages and standard deviations are computed over 10 independent samples. **C** Average computation time for different numbers of neurons $N$ and trials $L = 55$, 100, 300, 550, with error bars indicating standard deviations. Computation was performed on a Dell PowerEdge R750 server with two Intel Xeon 2.4 GHz CPUs (76 cores/152 threads).

which prevents it from capturing higher-order dependencies. Reproducing widespread spike-count histograms in large populations is known to require interactions of all orders[59]. By contrast, the pairwise model concentrates probability mass on only up to two points in the limit of large $N$, often overestimating the tail because it neglects the higher-order interactions that generate sparse, heavy-tailed distributions. The mismatch in model architectures is also apparent in their activation functions (Fig. 5B). The alternating-shrinking higher-order interaction model exhibits a supra-linear activation function due to the nonlinear integration of synaptic inputs (Eq. (71) in Methods). In contrast, the kinetic Ising model employs the classical logistic activation function with a linear sum of synaptic inputs (Eq. (1)).

In addition, an equally profound architectural limitation lies in the assumption of conditional independence, which enforces synchronous updates across neurons within each step. Gibbs sampling, by contrast, uses sequential (or randomly ordered) updates that guarantee detailed balance and allow neurons to incorporate the most recent changes, enabling activity to propagate within a sweep and generate synchronous states. Because the kinetic Ising model updates all neurons simultaneously from the previous state, it lacks this recruitment mechanism and consequently fails to drive synchronous activity appropriately.

The results highlight that caution is warranted in applying the kinetic Ising framework: although it offers a tractable statistical description, its simplifying assumptions constrain the neural dynamics it can represent. In particular, entropy flow estimates should be

regarded as quantities defined under the pairwise and synchronous-update assumptions.

## Mouse V1 neurons: experimental design and data description
Having confirmed the applicability of our methods using simulation data, we next applied the state-space kinetic Ising model to empirical data obtained from mice exposed to visual stimuli and estimated its entropy flow.

In this study, we analyzed the Allen Brain Observatory: Visual Behavior Neuropixels dataset provided by the Allen Institute for Brain Science, which contains large-scale recordings of neural spiking activity of mouse brains during the visual change detection task (See refs. 60–62 for analyses using this data set). The task is designed to analyze the effect of novelty and familiarity of the stimulus on neural responses. One of two image sets (G and H) was presented to animals at the training/habituation and recording sessions with different orders. The G and H image sets contain 8 natural images. We analyzed the recordings of 37 mice available from the Allen dataset, which were exposed to stimulus G in the recording sessions (either day 1 or 2) whereas the same stimulus G was used in the training and habituation sessions prior to the recording sessions (i.e., the case in which G is familiar).

The neural activities were recorded under two distinct conditions, in which the mice were either actively or passively performing the task under the same set of images. The active condition involved the mice performing a go/no-go change detection task, where they earned a

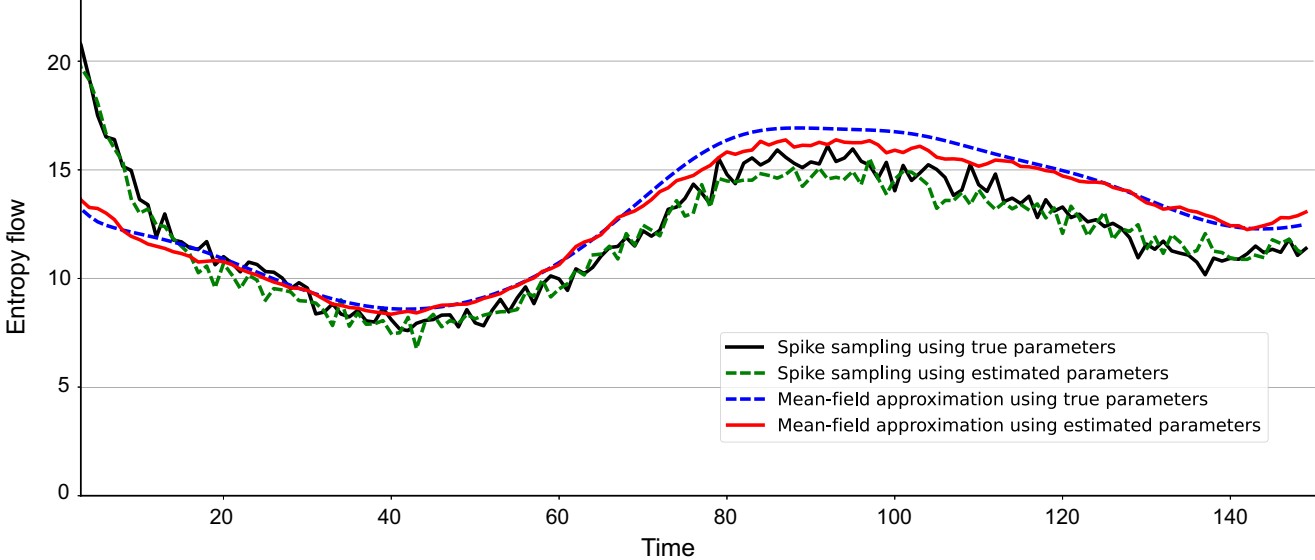

**Fig. 4 | Comparison of entropy flow estimation methods.** Entropy flows estimated using four different approaches: Sampling method with true parameters $\boldsymbol{\theta}_t$ (solid black); sampling method with estimated parameters $\boldsymbol{\theta}_{t|T}$ (dashed green); mean-field method with true parameters $\boldsymbol{\theta}_t$ (dashed blue); and mean-field method with estimated parameters $\boldsymbol{\theta}_{t|T}$ (solid red).

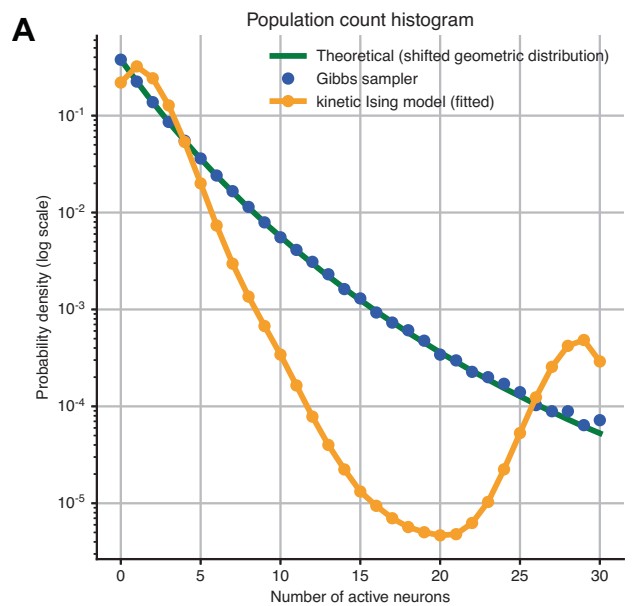

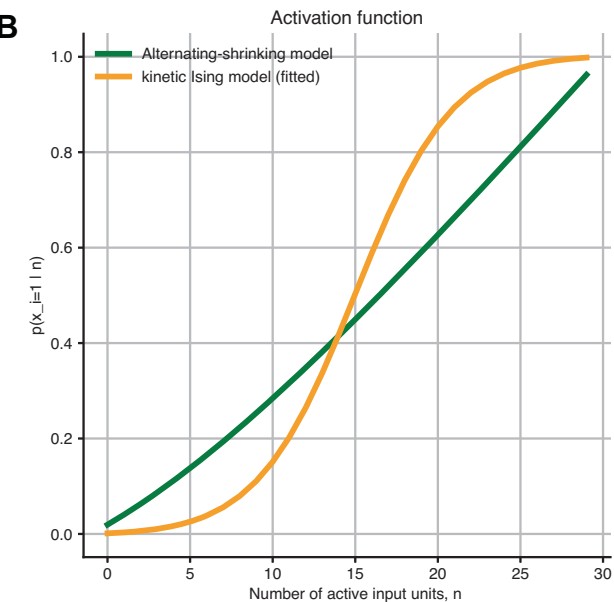

**Fig. 5 | Analysis of the model misspecification. A** The population spike count histogram of $N = 30$ neurons following the shifted-geometric model with a sparseness parameter $f = 20$ and $\tau = 0.8$ (empirical distribution obtained by the Gibbs sampling in blue circle; theoretical probabilities in green). The yellow line represents a distribution obtained from the state-space kinetic Ising model fitted to the Gibbs sampling data. **B** The activation function of the shifted-geometric model with $f = 20$ and $\tau = 0.8$ (green) and that of the kinetic Ising model (yellow) using the average of the fitted field and coupling parameters.

water reward upon detection of a change in the visual stimulus, measured by licking behavior. Each of the 8 stimuli was presented for 250 ms, followed by a 500 ms interstimulus interval (gray screen), repeating for one hour while mice actively engaged in the task for reward. In contrast, the passive condition involved replaying the same visual stimuli used in the active condition but without providing any rewards or access to the lick port. In this study, we analyzed recordings with images labeled im036_r, im012_r, and im115_r, which are used in the training session and classified as Familiar, and compared neural responses under the active and passive conditions. We used all presentations of the images equally and treated one presentation as a trial.

We selected neurons in the V1 area for analysis. For each mouse, we analyzed the simultaneous activity of neurons during a 750-ms period following the image onset. Although the number of trials varies across mice, the mean trial count was 566 with 356 and 652 as the minimum and maximum number of trials, respectively, for the case of an image im036_r.

**Mouse V1 neurons: an exemplary result from a single mouse**
We constructed binary sequences using a 10 ms bin, which resulted in $T = 75$ time bins. Here, we focused on the analysis of im036_r. Figure 6A (Left) shows the spike-rate averaged over neurons at each time under

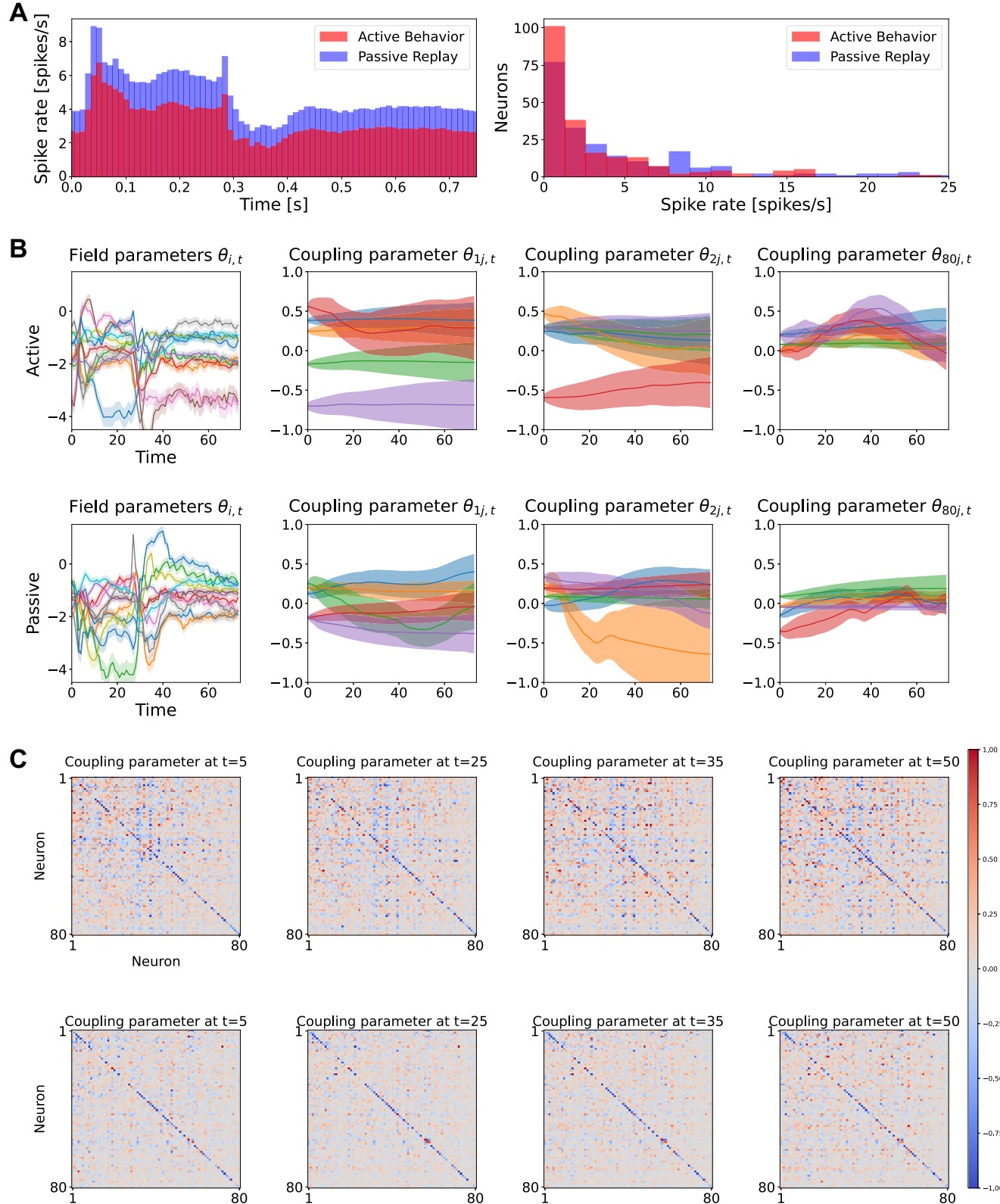

**Fig. 6 | Estimated neural dynamics under the active and passive conditions for mouse 574078. A** Spike-rate dynamics and distributions. (Left) Spike-rate averaged across neurons and trials. (Right) Spike-rate distributions of all recorded neurons. **B** Smoothed time-dependent parameters $\boldsymbol{\theta}_{t|T}$ of the kinetic Ising model for the active (top) and passive (bottom) conditions. The first column shows the field $\theta_{i,t}$ (one trace per neuron), and the next three columns show the incoming couplings $\theta_{ij,t}$ for $i = 1, 2, 80$. Solid lines are MAP estimates and shaded areas indicate $\pm 1$ SD (i.e., 68% credible bands) computed from the diagonal of the posterior covariance. For each $i$, couplings were first screened within the analysis window (bins 21–75) and retained if their credible interval excluded zero at least once in the window (self-couplings excluded). From the retained set, we display the first five couplings per $i$, ordered by ascending $j$ label for readability. **C** Estimated couplings at $t = 5, 25, 35, 50$ under the active (top) and passive (bottom) conditions.

the active and passive conditions (population-average spike rate) from an exemplary mouse (574078). The overall temporal profiles were similar across the active and passive conditions. In both conditions, the population exhibited higher mean spike rates during the stimulus presentation period (0–250 ms) than the post-stimulus period (250–750 ms). However, their magnitudes significantly differed across the conditions. The passive condition (blue) showed consistently higher spiking probabilities than the active condition (red) throughout the stimulus and post-stimulus periods. In agreement with the population-average spike-rate dynamics, time-averaged spike rates of individual neurons exhibited a sparser distribution during the active condition compared to the passive condition (Fig. 6A, Right).

We then applied the state-space kinetic Ising model to the binary activities of these neurons. For this goal, we selected the top $N = 80$ neurons with the highest spike rates. The estimated dynamics of the field and coupling parameters exhibited variations in both active and passive conditions (Fig. 6B). Notably, the field parameters $\theta_{i,t}$ (the first column) follow the dynamics of the mean spike rate of the population with significant fluctuations. On the contrary, the dynamics of the coupling parameters $\theta_{ij,t}$ exhibited smoother transitions. To clarify the dynamics of the couplings, we show them in the matrix form at specific time points, $t = 5, 25, 35, 50$ (Fig. 6C). The neurons are indexed in the ascending order of the average firing rates. The top and bottom rows show the results of the active and passive conditions, respectively. Coupling strength is indicated by graded color, with red and blue representing positive and negative values, respectively. The results show that (i) the couplings exhibit significant variations with positive and negative values; (ii) the variations are stronger in the active condition than in the passive condition; (iii) the diagonal components of the couplings (self-correlations) mostly display negative correlations.

To corroborate the above observations, we performed the same analysis on the trial-shuffled data (Supplementary Fig. S1). The analysis of trial-shuffled data reveals bias and variance in estimation under the assumption of neuronal independence. The result shows a significant reduction in the magnitude and variability of the couplings, whereas self-couplings remained unchanged (note that the self-coupling remains after trial-shuffling). However, non-zero couplings persisted with stronger variations in the active condition than in the passive condition, reflecting sampling fluctuations due to the lower firing rates in the active condition. These findings indicate that the parameters observed in Fig. 6B, C include estimation noise, necessitating statistical analyses to confirm their significance.

## Mouse V1 neurons: population analysis across mice

We assessed key features identified in the exemplary mouse (Fig. 6) across all mice by comparing them with trial-shuffled data.

First, the firing rate profiles with reduced activity in the active conditions found in Fig. 6A were consistently observed across all mice with a few exceptions (Supplementary Fig. S2 and S3). We compared the mean and sparsity of the firing rate distributions of individual neurons between the two conditions across all 37 mice (Supplementary Fig. S4). Sparsity of a non-negative firing rate distribution was quantified by the coefficient of variation (CV)[63]. The V1 neurons exhibited diminished and sparser firing rate distributions in the active condition than in the passive condition, as confirmed by the reduced mean spike rates ($p = 1.556 \times 10^{-8}$, Wilcoxon signed-rank test) and increased CVs ($p = 8.35 \times 10^{-8}$, Wilcoxon signed-rank test).

Next, we assessed key statistical features of the estimated parameters of the state-space kinetic Ising model. Figure 7A–C illustrates these features for an exemplary mouse (574078). Figure 7A, B shows distributions of time-averaged fields $\theta_{i,t}$ and couplings $\theta_{ij,t}$ under the active and passive conditions, while Figure 7C shows a scatter plot of time-averaged reciprocal couplings $\theta_{ij,t}$ vs $\theta_{ji,t}$ to evaluate coupling

asymmetry. In the active condition, the medians of field parameters decreased, reflecting reduced firing rates, while the medians of couplings remained near zero in both conditions. Field and coupling parameter variances increased, and coupling asymmetry strengthened in the active condition. These trends were consistent across all mice (Fig. 7D–F). These characteristics represent key aspects of neural dynamics that are closely related to entropy flow, although they are not entirely independent of each other.

While increased parameter variabilities and coupling asymmetry were observed under the active condition, they may be influenced by the lower neuronal activity. To examine this, we compared results with trial-shuffled data across all mice. Figure 7G–I shows field and coupling variances in both conditions, adjusted by subtracting shuffled data values for each mouse. Notably, observed values in both active and passive conditions were significantly higher than shuffled data: $p = 2.91 \times 10^{-11}$ (active), $p = 1.103 \times 10^{-7}$ (passive) for fields, $p = 4.676 \times 10^{-8}$ (active), $p = 1.455 \times 10^{-11}$ (passive) for couplings (Wilcoxon signed-rank test). Note that the observed significant heterogeneity in the field parameters is likely associated with the coupling heterogeneity. These results confirm that the variability observed in active or passive conditions is not explained by noise couplings. The coupling asymmetry was higher than shuffled results only for the active condition ($p = 1.185 \times 10^{-5}$(active) and $p = 0.1287$ (passive) for asymmetry).

Comparisons of these significant changes of the parameter variability (i.e., shuffled results subtracted) between the active and passive conditions showed significantly greater values in the active condition ($p = 8.273 \times 10^{-4}$ for fields, $p = 6.421 \times 10^{-4}$ for couplings, Wilcoxon signed-rank test, Fig. 7G, H), indicating greater variabilities in both field and coupling parameters during active behavior. A similar analysis of the mean couplings across mice revealed slightly but significantly larger values under the active condition (Supplementary Fig. S5). In contrast, coupling asymmetry showed no significant difference ($p = 0.1287$, Wilcoxon signed-rank test, Fig. 7I). The lack of statistically discernible change in asymmetry in the effective couplings accords with the use of the proposed mean-field method for comparing the coupling effect, which primarily arises from variability change. These findings validate enhanced parameter variability in the sparse neuronal activity during active engagement.

## Mouse V1 neurons: entropy flow dynamics

Using the estimated parameters of the state-space kinetic Ising model, we computed entropy flow dynamics. Figure 8A shows the time-varying entropy flow of a representative mouse (574078) under the active and passive conditions (red and blue solid lines, respectively). In both cases, transient increases in entropy flow coincided with declines in the mean population spike rate (dashed lines). Similar patterns appeared across all mice analyzed (Supplementary Fig. S6). These increases align with the second law, indicating that greater entropy dissipation is required when the system is transitioning to a lower entropy state, characterized by reduced firing rates.

The entropy flow time courses for this mouse showed no clear differences between the active and passive conditions. To assess population-level effects, we analyzed all 37 mice and computed total entropy flow across time bins for each condition (Fig. 8B). The comparison revealed significantly lower total entropy flow in the active condition ($p = 0.01159$, Wilcoxon signed-rank test). Note that neurons exhibited reduced firing rates (Supplementary Fig. S3) and increased parameter variability (Fig. 7D, E) during the active condition.

To isolate the effect of couplings, we compared the observed total entropy flows with shuffled data results (Fig. 8C). The estimated entropy flow for shuffled data includes the impact of firing rate dynamics and estimation error on couplings from other neurons;

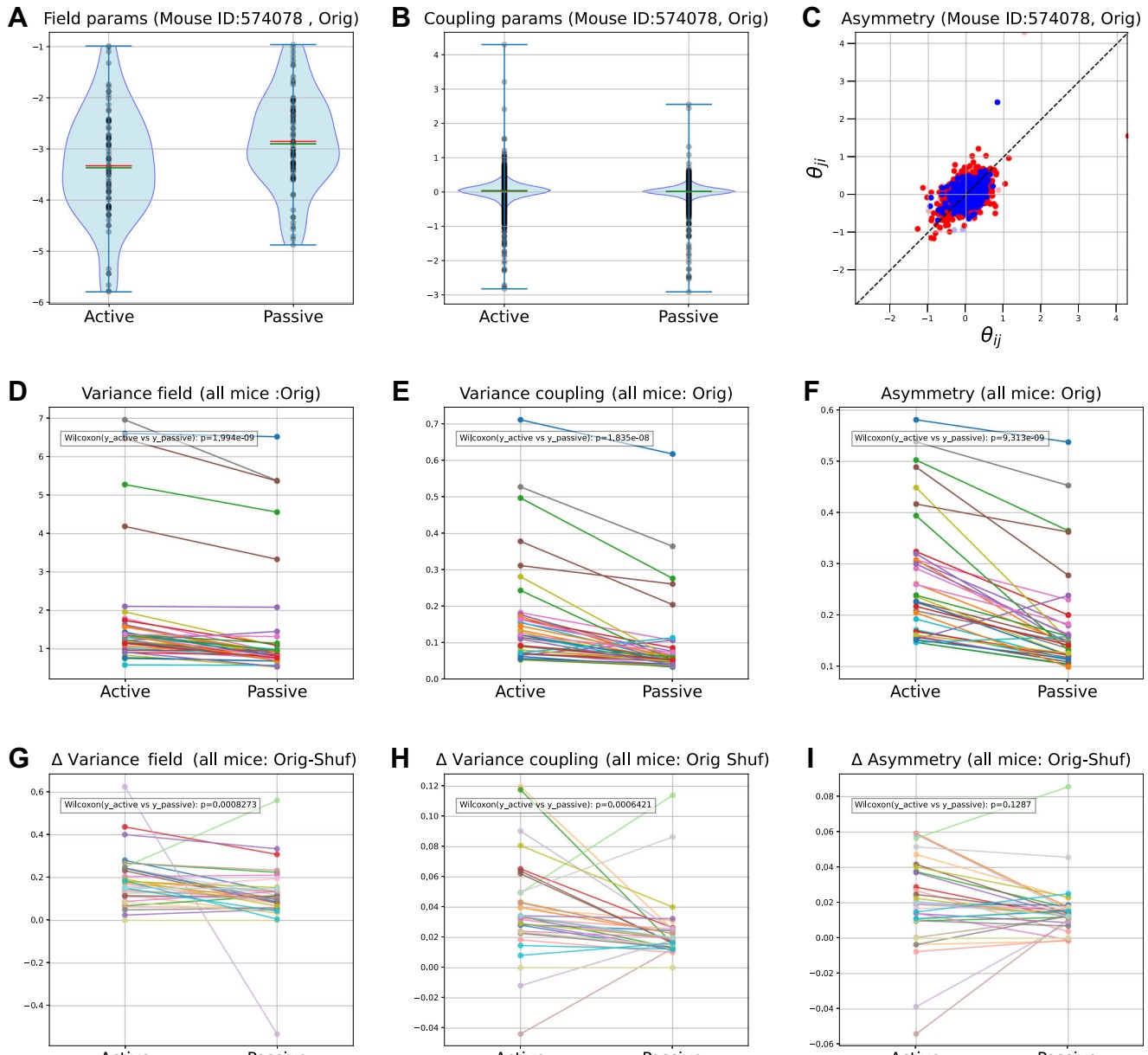

**Fig. 7 | Variability of estimated model parameters. A, B** Distributions of the time-averaged field values $\bar{\theta}_i$ (A) and the time-averaged couplings $\bar{\theta}_{ij}$ (B) for mouse 574078 under the active and passive conditions. The shaded violin plots depict kernel-density estimates of the empirical distributions; gray dots are the underlying observations (one dot per neuron in **A**, one per coupling in **B**). Short horizontal caps at the top and bottom indicate the sample maximum and minimum, respectively. Horizontal red bars mark the mean, while horizontal green bars mark the median. **C:** Scatter plots of coupling strength of reciprocal pairs under the active (red) and passive (blue) conditions for mouse 574078. The coupling asymmetries were 0.147 (active) and 0.105 (passive). The asymmetry was assessed by the average absolute difference of the reciprocal couplings $\langle|\bar{\theta}_{ij} - \bar{\theta}_{ji}|\rangle_{ij}$, where $\bar{\theta}_{ij}$ indicates the time-average of $\theta_{ij,t}$ and $\langle\cdot\rangle_{ij}$ refers to the average over the combinations of $i, j$. **D–F** Group-level (all mice) comparisons for the original dataset: field variance (D), coupling variance (E), and coupling asymmetry (F). **G–I:** Plots analogous to (**D–F**) for shuffle-subtracted parameter variances and coupling asymmetry. Each subplot of (**D–I**) contains the $p$-values of Wilcoxon signed-rank tests for the active vs. passive conditions.

therefore, subtracting shuffling results from observed entropy flow isolates contributions of couplings among different neurons beyond the sampling fluctuation. Positive values of the shuffle-subtracted total entropy flow in both conditions indicate that the couplings caused a significant entropy flow increase ($p = 1.455 \times 10^{-11}$ for active, $p = 1.455 \times 10^{-11}$ for passive, Wilcoxon signed-rank test). These shuffle-subtracted entropy flows behave in agreement with the theoretical prediction by the Sherrington-Kirkpatrick model[37]. In the active condition, the increased coupling variability (and asymmetry) from the shuffle-subtracted values were positively correlated with the shuffle-subtracted entropy flows, while the increased field heterogeneity was

negatively correlated (Supplementary Fig. S7A–C). These effects disappeared in the passive condition, possibly due to small changes in the variabilities and asymmetry introduced by shuffling (Supplementary Fig. S7D–F).

We analyzed the differences in coupling-related entropy flows between the active and passive conditions for all mice (Fig. 8C). The result shows no significant difference between the two conditions ($p = 0.1448$, Wilcoxon signed-rank test). However, coupling-related entropy flows of indistinguishable magnitude emerged under distinct neural activity states: sparser, lower activity with increased variability in field and coupling parameters in the active condition; and less

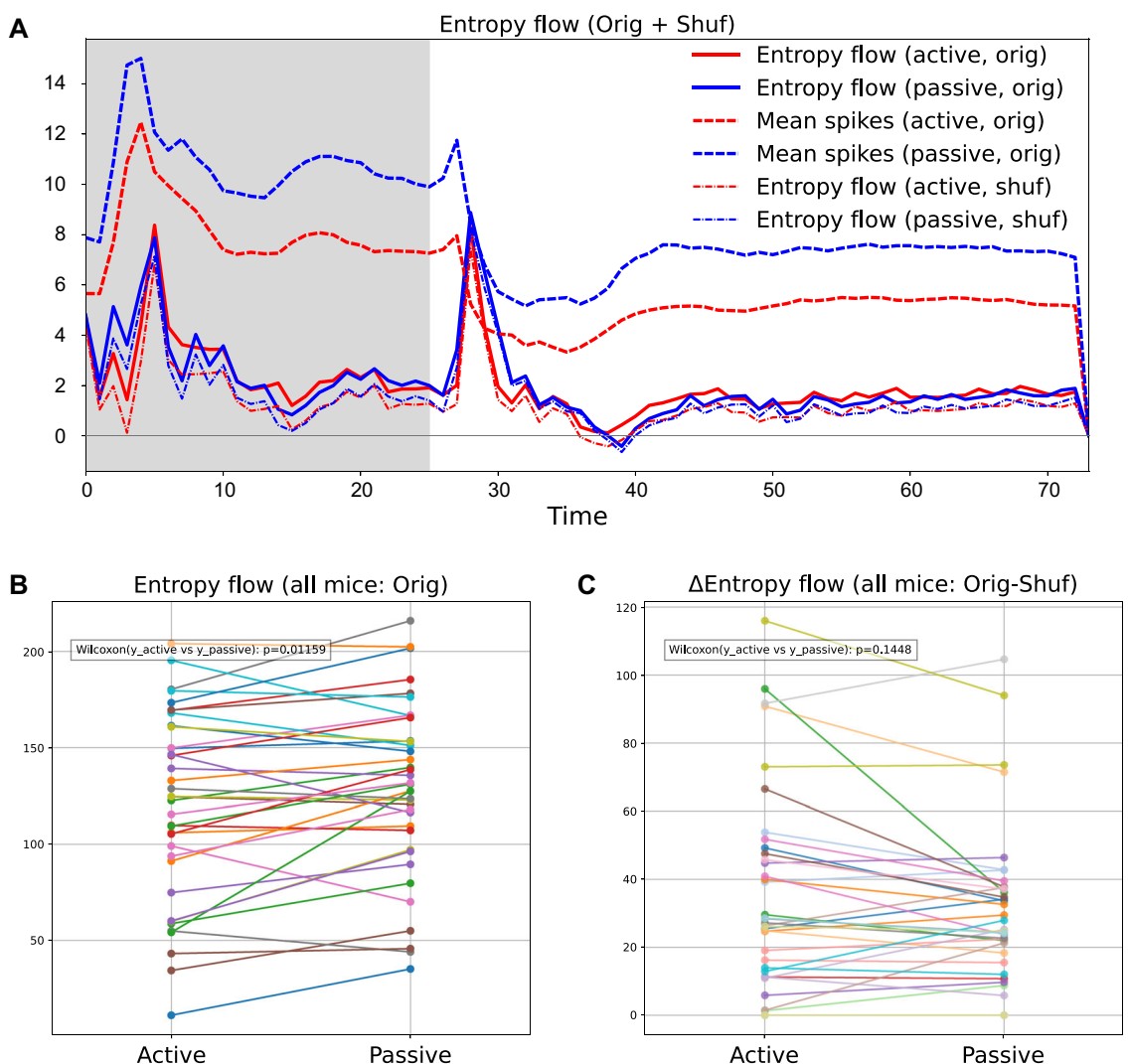

**Fig. 8 | Estimated entropy flow dynamics. A** Time courses of entropy flow for the original (solid lines) and shuffled data (dash-dot lines) under the active (red) and passive (blue) conditions. The dashed lines show the corresponding population-averaged spike rates. **B** Total entropy flows summed across all time bins for each mouse under active and passive conditions. Data from the same mouse are connected by a line. **C** Shuffle-subtracted total entropy flow (original - shuffle), shown for each mouse under the active and passive conditions.

sparse, higher activity with reduced variability in the passive condition. Thus, coupled with the previous results, this result indicates that the greater coupling variability in the active condition led to increased total entropy flow, making it comparable to the passive condition despite significantly sparser firing rate distributions. Consistent with this view, a recent study by Aguilera et al. using this dataset complementarily reported that a lower bound on entropy production, derived under steady-state assumptions using a variational framework, was higher in the active condition when normalized per spike[13].

**Mouse V1 neurons: model-based perturbation analysis**
To further elucidate the difference in the estimated entropy flow in active and passive conditions, we performed a model-based perturbation analysis by rescaling the fitted model parameters as $\theta \to \beta\theta$ and computing the resulting entropy flow to assess its sensitivity to parameter perturbation. An example result from mouse 574078 (Fig. 9A) shows that the entropy flows during stimulus presentation and waiting (gray image) periods exhibited distinct behaviors in response to the rescaling. The transient increases in entropy flow caused by firing rate reduction after stimulus onset and offset persisted as the scaling parameter $\beta$ increased. In contrast, we observed that the entropy flow

peaked at $\beta < 1$ during the waiting period, where the neural activity is relatively stationary.

Both forward and reverse conditional entropies ($\sigma_t^{\text{forward}}$ and $\sigma_t^{\text{backward}}$ in Eq. (12)) decreased with increasing $\beta$ during the waiting period (Fig. 9B, C), indicating that both processes became more deterministic. This trend suggests that, as $\beta$ increases, the system transitions from a disordered phase toward a ferromagnetic phase, rather than into a quasi-chaotic regime[43]. Thus, these results indicate that the subsampled neural population during this period operates in a subcritical regime.

By subtracting the entropy flow estimated from trial-shuffled data, which preserves only firing rate dynamics, we confirmed that the two bands of increased entropy flow associated with stimulus presentation are attributable to firing rate changes, whereas the increase at $\beta < 1$ arises from interactions, since the former disappeared but the latter persisted after shuffle subtraction (Fig. 9D). The interaction-driven entropy flow revealed by parameter scaling was stronger during the active condition than the passive condition (Fig. 9D, E), a result confirmed across all mice (Fig. 9F). Notably, the previous analysis at $\beta = 1$ showed no difference in shuffle-subtracted (i.e., interaction-driven) entropy flow between the two conditions (Fig. 8C). Thus, the

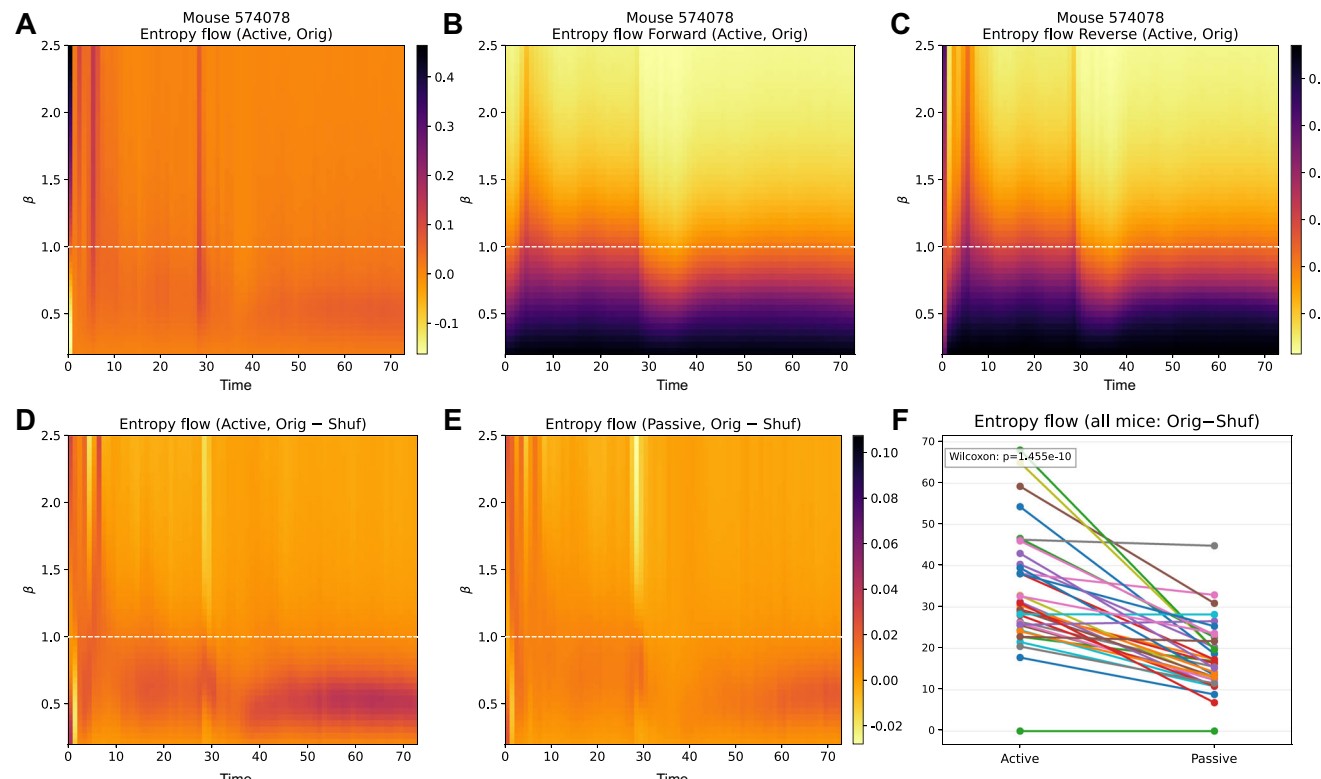

**Fig. 9 | Model-based perturbation analysis. A** Entropy flow $\sigma_t^{\text{flow}}$ of mouse 574078 in the active condition, computed after rescaling the fitted parameters as $\boldsymbol{\theta} \to \beta\boldsymbol{\theta}$. The dashed line indicates $\beta = 1$. **B** Forward entropy flow, $\sigma_t^{\text{forward}}$. **C** Backward entropy flow, $\sigma_t^{\text{backward}}$. **D** Shuffle-subtracted entropy flow in the active condition, isolating the contribution of interactions beyond firing rate dynamics. Entropy flow driven by interactions peaks at $\beta < 1$. **E** Shuffle-subtracted entropy flow in the passive condition. **F** Comparison of shuffle-subtracted entropy flow between active and passive conditions across all mice, showing significantly higher values in the active condition (Wilcoxon test, $p = 1.455 \times 10^{-10}$). Shuffle-subtracted entropy flow is obtained over a low-gain range $\beta \in [0.2, 1.0]$ and across all bins. Lines connect active (left) to passive (right) for each mouse.

model-based perturbation analysis uncovered differences in entropy flow between active and passive states that were not apparent at $\beta = 1$.

## Mouse V1 neurons: entropy flow and behavioral performance

Finally, we investigated the relationship between neural dynamics and behavioral performance across individual mice. We quantified task performance by the sensitivity index $d'$ (mean d-prime) defined as the difference between the z-transformed hit and false-alarm rates (Supplementary Note 4). In the following analyses, we extended the analysis to include two additional images (im012_r and im115_r).

First, we examined how sparseness, assessed from individual neurons' activity rates, relates to behavioral outcomes. As shown in Supplementary Fig. S3, neuronal activities were significantly reduced under active conditions, accompanied by increased sparsity of firing rate distributions. To further characterize this effect, we examined whether the reduction was uniform across neurons or driven by a subset of neurons by computing the skewness of the firing rate difference between active and passive conditions (Fig. 10A). A uniform reduction results in a skewness of zero, whereas negative skewness indicates that only some neurons decreased their activity, reflecting the sparsification. We found that this sparsification index was significantly correlated with behavioral performance measured by the d-prime, indicating that task engagement is reflected in changes in sparsity quantified at the level of individual neurons' activity rates (Fig. 10B).

Having established the link between activity-rate sparsity and behavior, we next turned to entropy flow to ask whether it provides additional explanatory power beyond rate changes alone. The variability of effective couplings was significantly higher during the active condition. To gain insight into the contributions of couplings to entropy flow, we computed the activity rate and mean-field entropy flow of individual neurons as a function of the mean and variability of their inputs (Fig. 10C, D). Theoretically, in the low-input and stationary regime, entropy flow increases with both higher mean input and greater variability (Eq. (20), background color in Fig. 10D). We observed that neurons receiving high mean input tended to have less variable inputs, whereas neurons with low mean input exhibited larger variability (colored circles). These results suggest that total entropy flow is shaped not only by high-input (typically high-firing) neurons but also by low-input neurons with high variability.

These patterns imply two sources of entropy flow: (i) mean-input-driven contributions that track high firing, and (ii) variability-driven contributions that can be substantial even at low firing. To focus more on the latter, we considered entropy flow per activity rate. This normalization reduces the direct dependence on the mean rate and makes variability-driven effects, particularly those arising in low-rate neurons, observable on an equal footing with high-rate effects. The shift in mean entropy flow per activity rate across individual neurons (active - passive) was significantly correlated with behavioral performance (Fig. 10E). Moreover, this correlation was weaker and non-significant for trial-shuffled data, indicating contributions from highly variable couplings during active conditions (Fig. 10F). This finding suggests that the thermodynamic cost per spiking activity is related to mouse performance, with couplings contributing in addition to activity-rate sparsity.

As an alternative explanation, behavioral performance could be related to entropy-flow changes concentrated in high-firing neurons. We therefore tested whether neurons with higher spike rates tended to increase entropy flow during active engagement in mice with higher task performance (Supplementary Note 5, Supplementary Fig. S8).

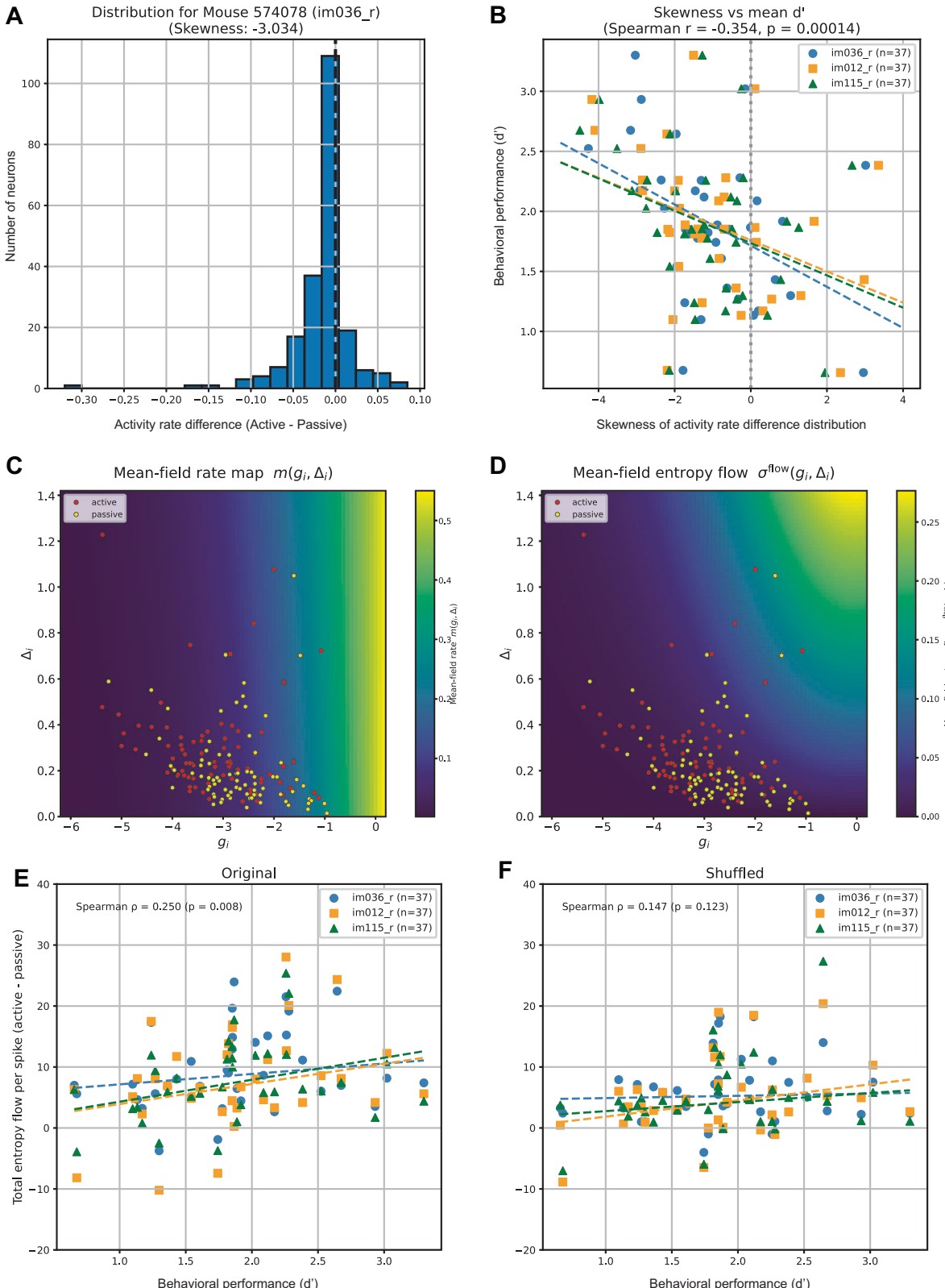

**Fig. 10 | Correlations between neural dynamics and behavioral performance.**
**A** A histogram of firing rate differences (active - passive) across neurons for mouse 574078 with an image im036_r. A skewness was used as a sparsification index. **B** The skewness vs behavioral performance (mean d-prime) for 37 mice with three images. **C** Mean-field rates of individual neurons (colored circles) as a function of time-average mean $g_i$ and variability $\Delta_i$ of their inputs for mouse 574078. The background color indicates the theoretical mean-field rate under the steady-state (Eq. (19)). **D** Mean-field entropy flow of individual neurons (colored circles) as a

function of time-average mean and variability of their inputs (mouse 574078). The background color shows theoretical entropy flow under the steady-state (Eq. (20)). **E** Entropy flow difference (active - passive) normalized by activity rate vs behavioral performance for 37 mice with three images. **F** Entropy flow difference normalized by activity rate obtained from trial-shuffled data vs behavioral performance. In E and F, there is one outlier mouse below −20 in the ordinate, which was included in the statistical analysis.

While this tendency correlated significantly with behavioral performance for one image, it was not significant for the other two images. We therefore infer that high-firing-based changes alone cannot consistently account for performance differences. Instead, the more robust association with entropy flow per activity rate supports a complementary role of variability-driven, coupling-mediated contributions—including those from low-rate neurons—in explaining behavioral performance.

## Discussion

This study presents a state-space kinetic Ising model for estimating nonstationary and nonequilibrium neural dynamics and introduces a mean-field method for entropy flow estimation. Through analysis of mouse V1 neurons, we identified distinct field and coupling distributions across behavioral conditions. These structural shifts influenced entropy flow compositions in V1 neurons, revealing correlations with behavioral performance.

To our knowledge, no inference methods have been proposed for time-dependent kinetic Ising models within the sequential Bayesian framework, which estimates parameters with uncertainty using optimized smoothness hyperparameters (see ref. 64 for a Bayesian approach in a stationary case). While parameter estimation has often been considered under time-dependent fields with fixed couplings[40,42] (see also refs. 65,66 for the equilibrium case), exceptions exist[41] that provide point estimates for time-varying couplings. These methods rely on mean-field equations relating equal-time and delayed correlations to coupling parameters, but estimating correlations at each time step is often infeasible in neuroscience data due to limited trial numbers in animal studies. Campajola et al.[67] proposed a point estimate of time-varying couplings using a score-driven method under the maximum likelihood principle, but assumed all fields and couplings were uniformly scaled by a single time-varying parameter. In contrast, our state-space framework accommodates heterogeneous parameter dynamics and employs sequential Bayesian estimation with optimized smoothness parameters. These innovations are crucial for uncovering parameter variability's impact on causal population dynamics and elucidating individual neurons' contributions.

Lower spike rates of V1 neurons observed during the active condition (see also ref. 60) contrast starkly with previous reports showing increased firing rates during active task engagement[68] or locomotion[69,70]. Nevertheless, the diminished spike rates found in the active condition (Fig. 6A and Supplementary Figs. S1, S2) are in agreement with sparse population activity in processing natural images in mouse V1 neurons[71,72]. Further, active engagement broadened distributions of field and coupling parameters, possibly reflecting stronger and more diverse inputs from hidden neurons[73,74]. These findings align with previously reported increased heterogeneous activities during the active condition and their correlation to behavioral performance[75]. The observed shift in cortical activity largely aligns with the effects of neuromodulators, such as acetylcholine (ACh)[76,77] and norepinephrine (NE)[78], which alter local circuit interactions and global activity patterns, thereby regulating transitions such as quiet-active, and inattentive-attentive states[79,80]. For example, Runfeldt et al.[76] demonstrated that spontaneous network events became sparser under ACh, as the probability of individual neurons participating in circuit activity was markedly reduced. In addition, ACh altered the temporal recruitment of neurons, delaying their activation relative to thalamic input and prolonging the window during which stereotyped activity propagated through local circuits. These findings indicate that ACh reorganizes cortical circuits into sparser and temporally extended modes of activity, potentially underlying the sparser population activity observed during task engagement and the stronger shift in entropy flow per spike in competent mice. However, we did not observe the previously reported decoupling of neuronal activity during active engagement (Supplementary Fig. S5), which may suggest the involvement of additional mechanisms beyond those described above.

In our analysis, the shift toward sparser activity during active engagement was significantly correlated with behavioral performance (Fig. 10B), consistent with sparse-coding theories that posit efficient representations using a few active neurons for natural images[81–83]. Moreover, mice with higher task performance exhibited greater entropy flow per spike during active compared with passive conditions (Fig. 10E), indicating that the capacity to form economical image representations via time–asymmetric causal activity is also linked to behavioral performance. Future work should determine whether this pattern reflects a direct computational mechanism or a secondary consequence of network state (e.g., attention or arousal). Importantly, the proposed method further yields testable predictions for information coding. For instance, if entropy flow per spike indeed relates to computation, then (i) neurons whose receptive fields match the presented image features should show selectively higher entropy flow per spike, or (ii) population decoding accuracy is expected to remain largely unchanged when the analysis is restricted to neurons with higher entropy flow per spike. Moreover, targeted pharmacological or optogenetic manipulations of neuromodulatory systems are predicted to induce systematic changes in entropy flow by modulating coupling variability, thereby altering coding efficiency. These predictions provide avenues to experimentally validate the computational role of entropy flow.

EEG, fMRI, and ECoG studies suggest that steady-state entropy production and related irreversibility metrics covary with consciousness level and cognitive load, and they reveal large-scale directed temporal structure[28–31,84–87]. For example, in human fMRI, violations of the fluctuation-dissipation theorem are larger during wakefulness than deep sleep, and larger during tasks than rest[85]. Arrow-of-time analyses likewise show stronger temporal asymmetry during tasks than rest and identify a cortical hierarchy of asymmetry[86]. Our state-space kinetic Ising model complements these steady-state, macroscopic approaches by estimating entropy flow directly from spiking data without assuming stationarity, potentially illuminating the lower-level mechanisms of mesoscopic/microscopic circuit dynamics. In parallel, equilibrium Ising and energy-landscape methods have been successfully applied to binarized neuroimaging and electrophysiological signals to characterize correlation structure and attractor basins of large-scale brain networks[88–91]. Our framework explicitly quantifies time-asymmetric entropy flow in nonstationary binary signals, complementing energy-landscape analyses of macroscopic stability with measures of time-dependent causal dynamics. In principle, our approach could be extended upward in scale to local field potentials (LFPs), multi-electrode arrays (MEAs), or coarse-grained EEG/ECoG recordings, enabling multiscale analysis of nonequilibrium dynamics from circuit to whole-brain levels.

In addition, our framework could be extended to analyze longer-term processes such as learning by treating time bins as trials within sessions and allowing parameters to vary across sessions, under the assumption of stationarity within each session. This would enable tracing learning trajectories of couplings among individual neurons when stable longitudinal recordings are available, an increasingly feasible scenario with recent advances in calcium imaging and electrophysiology[92,93]. However, the state-space method still faces limits in computational time and scale, constraining its use for large-scale signals. Future improvements through parallelization, optimized algorithms, and refined mean-field approaches could extend its applicability and enhance entropy flow estimation.

The kinetic Ising-based framework should also be viewed in light of its theoretical limitations. While analytically tractable, it imposes

strong assumptions—namely, pairwise couplings and conditional independence—that simplify neural dynamics but restrict interpretability. Our model misspecification analysis (Fig. 5) showed that reproducing the heavy-tailed spike-count statistics observed in real populations requires higher-order interactions; neglecting these leads to systematic biases, particularly in the tails. Likewise, synchronous updates imposed by conditional independence obscure cascade-like recruitment within bins in experimental data, leading to bin-size-dependent distortions: large bins capture heavy tails by merging cascades, which the model fails to represent, while small bins preserve fine-scale cascades, but the model misses slower interactions distant in time. These limitations motivate extensions beyond the synchronous pairwise framework. The generalized linear models (GLMs) and related point-process models provide a natural asynchronous alternative with longer history-dependency, since spikes are modeled in fine-grained bins or continuous time and influence others through coupling kernels. However, entropy flow in such history-dependent systems requires full path probabilities, making estimation challenging.

More broadly, fitted couplings and entropy flow should be regarded as statistical summaries of nonequilibrium dynamics, not direct measures of synaptic connectivity or mechanism. Future work must relax these constraints—by permitting asynchronous updates, incorporating higher-order dependencies, and developing principled estimators of entropy flow in non-Markovian settings—while remaining clear about the limits of inference when bridging statistical abstractions with physiology. For example, the alternating-shrinking higher-order interaction model (Eq. (63)) could be extended to include asymmetric couplings, potentially with asynchronous updates in a continuous-time limit.

In summary, by developing a state-space kinetic Ising model that accounts for both nonstationary and nonequilibrium properties, we have demonstrated how task engagement modulates neuronal firing activity and coupling diversity. Our approach incorporates time-varying entropy flow estimation, revealing that time-asymmetric, irreversible activity emerges within sparsely active populations during task engagement—an effect correlated with the mouse's behavioral performance. These findings underscore the utility of our approach, offering new insights into the thermodynamic underpinnings of neural computation.

## Methods

### Estimating time-varying parameters of the kinetic Ising model

We summarize the expectation-maximization algorithm for estimating the state-space kinetic Ising model with optimized hyperparameters. See Supplementary Note 1 for more details.

E-step: Given the hyperparameters $\mathbf{w}$, we obtain the estimate of the state $\boldsymbol{\theta}_t$ given all the data available. When estimating the parameters $\boldsymbol{\theta}_t^i$ ($t = 0, 1, \ldots, T, i = 1, \ldots, N$) from the spike data $\mathbf{x}_t$ ($t = 0, 1, \ldots, T$), we first obtain the filter density by sequentially applying the Bayes theorem:

$$p(\boldsymbol{\theta}_t | \mathbf{x}_{0:t}, \mathbf{w}) = \frac{p(\mathbf{x}_t | \boldsymbol{\theta}_t, \mathbf{x}_{0:t-1}, \mathbf{w}) p(\boldsymbol{\theta}_t | \mathbf{x}_{0:t-1}, \mathbf{w})}{p(\mathbf{x}_t | \mathbf{x}_{0:t-1}, \mathbf{w})}. \tag{22}$$

Here, the one-step prediction density is computed using the Chapman-Kolmogorov equation:

$$p(\boldsymbol{\theta}_t | \mathbf{x}_{0:t-1}, \mathbf{w}) = \prod_{i=1}^{N} \int p(\boldsymbol{\theta}_t^i | \boldsymbol{\theta}_{t-1}^i, \mathbf{Q}^i) p(\boldsymbol{\theta}_{t-1}^i | \mathbf{x}_{t-1}) d\boldsymbol{\theta}_{t-1}^i. \tag{23}$$

By assuming that the filter density for the $i$th neuron at the previous time step $t-1$ is given by the Gaussian distribution with mean $\boldsymbol{\theta}_{t-1|t-1}^i$ and covariance $\mathbf{W}_{t-1|t-1}^i$, the one-step prediction density becomes the Gaussian distribution whose mean $\boldsymbol{\theta}_{t|t-1}^i$ and covariance $\mathbf{W}_{t|t-1}^i$ are

given by

$$\boldsymbol{\theta}_{t|t-1}^i = \boldsymbol{\theta}_{t-1|t-1}^i, \tag{24}$$

$$\mathbf{W}_{t|t-1}^i = \mathbf{W}_{t-1|t-1}^i + \mathbf{Q}^i, \tag{25}$$

with $\boldsymbol{\theta}_{1|0}^i = \boldsymbol{\mu}^i$ and $\mathbf{W}_{1|0}^i = \boldsymbol{\Sigma}^i$ being the hyperparameters of the initial Gaussian distribution, $p(\boldsymbol{\theta}_1^i | \boldsymbol{\mu}^i, \boldsymbol{\Sigma}^i)$. Then, the filter density is given as

$$p(\boldsymbol{\theta}_t | \mathbf{x}_{0:t}, \mathbf{w}) = \prod_{i=1}^{N} p(\boldsymbol{\theta}_t^i | \mathbf{x}_{0:t}, \mathbf{w})$$
$$\propto \prod_{i=1}^{N} \prod_{l=1}^{L} \exp\left[ \theta_{i,t} x_{i,t}^l + \sum_{j=1}^{N} \theta_{ij,t} x_{it}^l x_{j,t-1}^l - \psi(\boldsymbol{\theta}_t^i, \mathbf{x}_{t-1}^l) \right]$$
$$\cdot \sum_{i=1}^{N} \exp\left[ -\frac{1}{2} (\boldsymbol{\theta}_t^i - \boldsymbol{\theta}_{t|t-1}^i)^\top (\mathbf{W}_{t|t-1}^i)^{-1} (\boldsymbol{\theta}_t^i - \boldsymbol{\theta}_{t|t-1}^i) \right]. \tag{26}$$

Since this filter density is a concave function with respect to $\boldsymbol{\theta}_t^i$ for each neuron, we apply the Laplace approximation independently to the filter densities of individual neurons and obtain the approximate Gaussian distributions, where the mean is approximated by the MAP estimate:

$$\boldsymbol{\theta}_{t|t}^i = \arg\max_{\boldsymbol{\theta}_t^i} \log p(\boldsymbol{\theta}_t^i | \mathbf{x}_{0:t}, \mathbf{w}), \tag{27}$$

for $i = 1, \ldots, N$, while the covariance is approximated using the Hessian as

$$\mathbf{W}_{t|t}^i = \left[ -\frac{\partial^2 \log p(\boldsymbol{\theta}_t^i | \mathbf{x}_{0:t}, \mathbf{w})}{\partial \boldsymbol{\theta}_t^i \partial \boldsymbol{\theta}_t^{i\top}} \bigg|_{\boldsymbol{\theta}_t^i = \boldsymbol{\theta}_{t|t}^i} \right]^{-1}$$
$$= \left[ \mathbf{G}(\boldsymbol{\theta}_{t|t}^i) + \left( \mathbf{W}_{t|t-1}^i \right)^{-1} \right]^{-1}, \tag{28}$$

where $\mathbf{G}(\boldsymbol{\theta}_t^i) \equiv \sum_{l=1}^{L} \frac{\partial^2 \psi(\boldsymbol{\theta}_t^i, \mathbf{x}_{t-1}^l)}{\partial \boldsymbol{\theta}_t^i \partial \boldsymbol{\theta}_t^i} \big|_{\boldsymbol{\theta}_t^i = \boldsymbol{\theta}_{t|t}^i}$ is the Fisher information matrix with respect to $\boldsymbol{\theta}_t^i$ computed for the kinetic Ising model over the trials. We computed the MAP estimate by the Newton-Raphson method utilizing the Hessian evaluated at a search point.

Next, we obtain the smoother density by recursively applying the formulae below. Because the filter density and state transitions are approximated by normal distributions, we follow the fixed-interval smoothing algorithm developed for the Gaussian distributions[94]. In this method, the smoothed mean and covariance are recursively obtained by the following equations:

$$\boldsymbol{\theta}_{t-1|T}^i = \boldsymbol{\theta}_{t-1|t-1}^i + \mathbf{A}_{t-1}^i (\boldsymbol{\theta}_{t|T}^i - \boldsymbol{\theta}_{t|t}^i), \tag{29}$$

$$\mathbf{W}_{t-1|T}^i = \mathbf{W}_{t-1|t-1}^i + \mathbf{A}_{t-1}^i (\mathbf{W}_{t|T}^i - \mathbf{W}_{t|t}^i) \mathbf{A}_{t-1}^{i\top}, \tag{30}$$

$$\mathbf{A}_{t-1}^i = \mathbf{W}_{t-1|t-1}^i \left( \mathbf{W}_{t|t-1}^i \right)^{-1}, \tag{31}$$

for $t = T, T-1, \ldots, 2$.

M-step: We optimize the hyperparameters given the smoothed posteriors. To optimize the hyperparameter $\mathbf{Q}^i$, we used the following update formula that maximizes the lower bound of the log marginal

likelihood:

$$
\mathbf{Q}^i = \frac{1}{T-1} \sum_{t=2}^{T} \Big[ (\boldsymbol{\theta}_{t|T}^i - \boldsymbol{\theta}_{t-1|T}^i)(\boldsymbol{\theta}_{t|T}^i - \boldsymbol{\theta}_{t-1|T}^i)^\top
$$
$$
+ \mathbf{W}_{t|T}^i - \mathbf{W}_{t-1,t|T}^i - \mathbf{W}_{t,t-1|T}^i + \mathbf{W}_{t-1|T}^i \Big].
$$
(32)

We compute the lag-one smoothing covariance matrix $\mathbf{W}_{t,t-1|T}^i$ following the method of De Jong and Mackinnon[95]: $\mathbf{W}_{t,t-1|T}^i = \mathbf{W}_{t|t}^i (\mathbf{W}_{t+1|t}^i)^{-1} \mathbf{W}_{t|T}^i$. We also note that the optimization of a diagonal of the form $\mathbf{Q}^i = \mathrm{diag}[\lambda_0^i, \ldots, \lambda_N^i]$ or $\mathbf{Q}^i = \lambda^i \mathbf{I}$ can be performed by taking diagonal and trace of the r.h.s of the equation above, respectively.

Similarly, we update $\boldsymbol{\Sigma}^i$ according to

$$
\boldsymbol{\Sigma}^i = \mathbf{W}_{1|T}^i + (\boldsymbol{\theta}_{1|T}^i - \boldsymbol{\mu})(\boldsymbol{\theta}_{1|T}^i - \boldsymbol{\mu})^\top.
$$
(33)

The convergence of the EM algorithm is assessed by computing the approximate log marginal likelihood function (Eq.(6)) using the Laplace approximation. Using the mean and covariance of the filter and one-step prediction densities, the approximate log marginal likelihood function for the hyperparameters $\mathbf{w}$ is obtained as

$$
\log p(\mathbf{x}_{0:T}|\mathbf{w}) = \log p(\mathbf{x}_0)
$$
$$
+ \sum_{t=1}^{T} \sum_{i=1}^{N} \Big[ \frac{1}{2} \log |\mathbf{W}_{t|t}^i| - \frac{1}{2} \log |\mathbf{W}_{t|t-1}^i| + q(\boldsymbol{\theta}_{t|t}^i) \Big].
$$
(34)

See Supplementary Note 1 for the derivations and the functional form of $q(\cdot)$.

### Mean-field approximation of the entropy flow

Here we extend the mean-field approximation method developed for the steady-state kinetic Ising model[43] to make it applicable to nonstationary systems.

First, $\sigma_t^{\mathrm{flow}}$ can be decomposed as follows by introducing the forward and backward conditional entropies:

$$
\sigma_t^{\mathrm{flow}} = - \sigma_t^{\mathrm{forward}} + \sigma_t^{\mathrm{backward}},
$$
(35)

where

$$
\sigma_t^{\mathrm{forward}} = - \sum_{\mathbf{x}_t, \mathbf{x}_{t-1}} p(\mathbf{x}_t, \mathbf{x}_{t-1}) \log p(\mathbf{x}_t|\mathbf{x}_{t-1}),
$$
(36)

$$
\sigma_t^{\mathrm{backward}} = - \sum_{\mathbf{x}_t, \mathbf{x}_{t-1}} p(\mathbf{x}_t, \mathbf{x}_{t-1}) \log p(\mathbf{x}_{t-1}|\mathbf{x}_t).
$$
(37)

We calculate these conditional entropies using the Gaussian approximation as follows.

We begin with approximating the forward conditional entropy as

$$
\sigma_t^{\mathrm{forward}} = - \sum_{\mathbf{x}_t, \mathbf{x}_{t-1}} p(\mathbf{x}_t|\mathbf{x}_{t-1}) p(\mathbf{x}_{t-1}) \log p(\mathbf{x}_t|\mathbf{x}_{t-1})
$$
$$
\simeq - \sum_{\mathbf{x}_{t-1}} Q(\mathbf{x}_{t-1}) \sum_{\mathbf{x}_t} p(\mathbf{x}_t|\mathbf{x}_{t-1}) \log p(\mathbf{x}_t|\mathbf{x}_{t-1}).
$$
(38)

Here we replaced $p(\mathbf{x}_{t-1})$ with an independent model $Q(\mathbf{x}_{t-1})$ defined as

$$
Q(\mathbf{x}_{t-1}) = \prod_i Q(x_{i,t-1}).
$$
(39)

The conditional probability is written as

$$
p(\mathbf{x}_t|\mathbf{x}_{t-1}) = \prod_i p(x_{i,t}|\mathbf{x}_{t-1}),
$$
(40)

where

$$
p(x_{i,t}|\mathbf{x}_{t-1}) = e^{x_{i,t} h_{i,t}(\mathbf{x}_{t-1}) - \psi(h_{i,t}(\mathbf{x}_t))},
$$
(41)

with

$$
h_{i,t}(\mathbf{x}_{t-1}) = \theta_{i,t} + \sum_j \theta_{ij,t} x_{j,t-1}.
$$
(42)

Here, we redefined the log normalization function $\psi$ as a function of $h_{i,t}(\mathbf{x}_t)$: $\psi(h_{i,t}(\mathbf{x}_t)) = \log(1 + e^{h_{i,t}(\mathbf{x}_t)})$.

Note that the expectation of $x_{i,t}$ is given by

$$
r(h_{i,t}(\mathbf{x}_{t-1})) = \sum_{x_{i,t}} x_{i,t} p(x_{i,t}|\mathbf{x}_{t-1})
$$
$$
= \frac{1}{1 + e^{-h_{i,t}(\mathbf{x}_{t-1})}}.
$$
(43)

Using $r(h_{i,t}(\mathbf{x}_{t-1}))$, we have

$$
p(x_{i,t}=1|\mathbf{x}_{t-1}) = r(h_{i,t}(\mathbf{x}_{t-1})),
$$
(44)

$$
p(x_{i,t}=0|\mathbf{x}_{t-1}) = 1 - r(h_{i,t}(\mathbf{x}_{t-1})).
$$
(45)

Then the forward conditional entropy becomes

$$
\sigma_t^{\mathrm{forward}} \simeq - \sum_{\mathbf{x}_{t-1}} Q(\mathbf{x}_{t-1}) \sum_i \sum_{x_{i,t}} p(x_{i,t}|\mathbf{x}_{t-1}) \log p(x_{i,t}|\mathbf{x}_{t-1})
$$
$$
= \sum_i \sum_{\mathbf{x}_{t-1}} Q(\mathbf{x}_{t-1}) \chi(h_{i,t}(\mathbf{x}_{t-1})),
$$
(46)

where

$$
\chi(h_{i,t}(\mathbf{x}_{t-1})) \equiv - \sum_{x_{i,t}} p(x_{i,t}|\mathbf{x}_{t-1}) \log p(x_{i,t}|\mathbf{x}_{t-1})
$$
$$
= - r(h_{i,t}(\mathbf{x}_{t-1})) \log r(h_{i,t}(\mathbf{x}_{t-1}))
$$
$$
- (1 - r(h_{i,t}(\mathbf{x}_{t-1}))) \log(1 - r(h_{i,t}(\mathbf{x}_{t-1})))
$$
$$
= - [r(h_{i,t}(\mathbf{x}_{t-1})) h_{i,t}(\mathbf{x}_{t-1}) - \psi(h_{i,t}(\mathbf{x}_t))]
$$
(47)

We approximate Eq. (46) by a Gaussian distribution based on the central limit theorem for a collection of independent binary signals. Specifically, by using $\mathcal{D}_z = \frac{dz}{\sqrt{2\pi}} \exp(-\frac{1}{2}z^2)$, the forward conditional entropy is approximated as

$$
\sigma_t^{\mathrm{forward}} \approx \sum_i \int \mathcal{D}_z \chi\Big(g_{i,t,t-1} + z\sqrt{\Delta_{i,t,t-1}}\Big),
$$
(48)

where $g_{i,t,t-1}$ and $\Delta_{i,t,t-1}$ are the mean and variance of $h_{i,t}(\mathbf{x}_{t-1})$ given by

$$
g_{i,t,t-1} = \theta_{i,t} + \sum_j \theta_{ij,t} m_{j,t-1},
$$
(49)

$$
\Delta_{i,t,t-1} = \sum_j \theta_{ij,t}^2 m_{j,t-1}(1 - m_{j,t-1}).
$$
(50)

Here, $m_{i,t}$ is the mean-field approximation of $x_{i,t}$ obtained by the Gaussian approximation method assuming independent activity of neurons at $t-1$:

$$
m_{i,t} = \sum_{\mathbf{x}_t, \mathbf{x}_{t-1}} x_{i,t} p(\mathbf{x}_t|\mathbf{x}_{t-1}) p(\mathbf{x}_{t-1})
$$
$$
= \sum_{\mathbf{x}_{t-1}} Q(\mathbf{x}_{t-1}) r(h_{i,t}(\mathbf{x}_{t-1})).
$$
(51)

Applying the Gaussian approximation to $h_{i,t}(\mathbf{x}_{t-1})$, $m_{i,t}$ is recursively computed as

$$m_{i,t} \approx \int \mathcal{D}z \, r\left(g_{i,t,t-1} + z\sqrt{\Delta_{i,t,t-1}}\right), \tag{52}$$

for $t = 1,\ldots,T$, using Eqs. (49) and (50), which are functions of $m_{i,t-1}$. Here $m_{i,1}$ was computed using nominal values of $m_{i,0}$ ($i = 1,\ldots,N$). In the simulation and empirical analyses, we used spiking probability averaged over all time steps and trials for each neuron as $m_{i,0}$.

Next, we approximate $\sigma_t^{\text{backward}}$. It is computed as

$$
\begin{aligned}
\sigma_t^{\text{backward}} &= -\sum_{\mathbf{x}_t, \mathbf{x}_{t-1}} p(\mathbf{x}_t, \mathbf{x}_{t-1}) \log p(\mathbf{x}_{t-1}|\mathbf{x}_t) \\
&= -\sum_{\mathbf{x}_t, \mathbf{x}_{t-1}} p(\mathbf{x}_t|\mathbf{x}_{t-1}) \sum_{\mathbf{x}_{t-2}} p(\mathbf{x}_{t-1}|\mathbf{x}_{t-2}) p(\mathbf{x}_{t-2}) \log p(\mathbf{x}_{t-1}|\mathbf{x}_t) \\
&= -\sum_{\mathbf{x}_{t-2}} \sum_{\mathbf{x}_{t-1}} p(\mathbf{x}_{t-1}|\mathbf{x}_{t-2}) p(\mathbf{x}_{t-2}) \\
&\quad \cdot \sum_{\mathbf{x}_t} p(\mathbf{x}_t|\mathbf{x}_{t-1}) \sum_i \left[ x_{i,t-1} h_{i,t}(\mathbf{x}_t) - \psi(h_{i,t}(\mathbf{x}_t)) \right].
\end{aligned}
\tag{53}
$$

We approximate the following probabilities by independent distributions:

$$p(\mathbf{x}_{t-2}) = Q(\mathbf{x}_{t-2}), \tag{54}$$

$$p(\mathbf{x}_t|\mathbf{x}_{t-1}) = Q(\mathbf{x}_t). \tag{55}$$

Using them, $\sigma_t^{\text{backward}}$ can be approximated as

$$
\begin{aligned}
\sigma_t^{\text{backward}} &\simeq -\sum_{\mathbf{x}_{t-2}} \sum_{\mathbf{x}_{t-1}} p(\mathbf{x}_{t-1}|\mathbf{x}_{t-2}) Q(\mathbf{x}_{t-2}) \\
&\quad \cdot \sum_{\mathbf{x}_t} Q(\mathbf{x}_t) \sum_i \left[ x_{i,t-1} h_{i,t}(\mathbf{x}_t) - \psi(h_{i,t}(\mathbf{x}_t)) \right] \\
&= -\sum_{\mathbf{x}_t} Q(\mathbf{x}_t) \sum_{\mathbf{x}_{t-2}} Q(\mathbf{x}_{t-2}) \\
&\quad \cdot \sum_i \left[ r(h_{i,t-1}(\mathbf{x}_{t-2})) h_{i,t}(\mathbf{x}_t) - \psi(h_{i,t}(\mathbf{x}_t)) \right] \\
&= -\sum_i \sum_{\mathbf{x}_t} Q(\mathbf{x}_t) \left[ m_{i,t-1} h_{i,t}(\mathbf{x}_t) - \psi(h_{i,t}(\mathbf{x}_t)) \right],
\end{aligned}
\tag{56}
$$

where we used Eq. (43) to obtain the second equality and Eq. (51) to obtain the last result. By defining

$$\phi_{i,t}(h_{i,t}(\mathbf{x}_t)) = -[m_{i,t-1} h_{i,t}(\mathbf{x}_t) - \psi(h_{i,t}(\mathbf{x}_t))], \tag{57}$$

the backward conditional entropy is obtained by the Gaussian integral:

$$
\begin{aligned}
\sigma_t^{\text{backward}} &= \sum_i \sum_{\mathbf{x}_t} Q(\mathbf{x}_t) \phi_{i,t}(h_{i,t}(\mathbf{x}_t)) \\
&\approx \sum_i \int \mathcal{D}z \, \phi_{i,t}\left(g_{i,t,t} + z\sqrt{\Delta_{i,t,t}}\right),
\end{aligned}
\tag{58}
$$

where

$$g_{i,t,t} \equiv \theta_{i,t} + \sum_j \theta_{ij,t} m_{j,t}, \tag{59}$$

$$\Delta_{i,t,t} = \sum_j \theta_{ij,t}^2 m_{j,t}(1 - m_{j,t}). \tag{60}$$

An alternative approach to obtain the backward conditional entropy is given in Supplementary Note 2.

Thus, the entropy flow is obtained as

$$
\begin{aligned}
\sigma_t^{\text{flow}} &= -\sigma_t^{\text{forward}} + \sigma_t^{\text{backward}} \\
&\approx \sum_i \int \mathcal{D}z \left[ -\chi\left(g_{i,t,t-1} + z\sqrt{\Delta_{i,t,t-1}}\right) \right. \\
&\quad \left. + \phi_{i,t}\left(g_{i,t,t} + z\sqrt{\Delta_{i,t,t}}\right) \right],
\end{aligned}
\tag{61}
$$

which allows us to examine the contributions of each neuron to the total entropy flow.

See also Supplementary Note 3 for the analytical expression of the entropy flow under steady-state conditions or for independent neurons.

## Generation of field and coupling parameters for simulation studies

We constructed time-varying field and coupling parameters, from which we generated the binary data. To ensure smooth temporal variations, each coupling parameter $\theta_{ij,t}$ was sampled from a Gaussian process of size $T$ with mean $\mu$ and covariance matrix defined by the squared exponential kernel

$$k(t,s) = k_0 \exp\left(-\frac{(t-s)^2}{2\tau^2}\right). \tag{62}$$

For the analysis of estimation error and computational time using different system sizes (Fig. 3), we used the scaling mean $\mu = 5/N$ and variance $k_0 = 10/N$, following the convention of the Sherrington-Kirkpatrick model. The characteristic length-scale was specified by $\tau = 30/\sqrt{N}$. Similarly, the external field parameters $\theta_{i,t}$ were independently sampled from the Gaussian process, using $\mu = -3$, $\tau = 50$, and $k_0 = 1$.

To obtain trajectories for the different system sizes, a single set of random values was generated for the maximum system size, and subsets of these values were used to examine the system size $N$. Specifically, for the coupling parameters, a global three-dimensional array was created with dimensions corresponding to the maximum number of neurons, time steps, and coupling connections. Similarly, for the field parameters, a two-dimensional array was generated, with dimensions corresponding to the maximum number of time steps and neurons. For a given neuron count $N$, the relevant subset of values was extracted from these precomputed arrays, ensuring that each $N$ used a subset of the values assigned to larger $N$. This hierarchical structure ensured that the seed for $N = 80$ encompassed all values used for smaller $N$, maintaining consistency across different system sizes. We evaluated the model's performance using this data set and repeated the procedure 10 times.

## Alternating-shrinking higher-order interaction model

To perform the analysis on fitting the kinetic Ising model to a mismatched model, we generated binary spike sequences using a nonlinearity that goes beyond linear synaptic summation and a logistic activation function, which therefore induces the higher-order interactions (HOIs) in the population activity. For this goal, we employed the recently proposed alternating-shrinking HOI model[58].

The model is a time-independent, homogeneous model including all orders of HOIs in the following form:

$$p(\mathbf{x}) = \frac{h\left(\sum_{i=1}^N x_i\right)}{Z} \exp\left[-f \sum_{j=1}^N (-1)^{j+1} C_j \left(\frac{\sum_i x_i}{N}\right)^j\right], \tag{63}$$

where $f$ is a sparsity parameter and $Z$ is the partition function. Let $n = \sum_{i=1}^N x_i$. $h(n)$ is an entropy-canceling base measure function defined

using the binomial coefficient:

$$h(n) = 1 \Big/ \binom{N}{n}.$$ (64)

The parameters $C_1, C_2, \ldots, C_N$ are the shrinking parameters, where $C_j = (\tau)^j$ with $0 < \tau < 1$ results in the shifted-geometric population spike-count distribution.

The population spike-count distribution is the probability distribution of $n$ active neurons in the binary patterns, which is given as

$$
\begin{aligned}
P(n) &= \binom{N}{n} p(x_1 = 1, \ldots, x_n = 1, x_{n+1} = 0, \ldots, x_N = 0) \\
&= \binom{N}{n} \frac{h(n)}{Z} \exp\left[-f \sum_{j=1}^{N} (-1)^{j+1} C_j \left(\frac{n}{N}\right)^j\right].
\end{aligned}
$$ (65)

This distribution was shown to be widespread due to the cancellation of the binomial term, and also sparse due to the alternating HOIs.

We performed Gibbs sampling from this distribution, which dictates the dynamics of a recurrent neural network with threshold-supralinear activation nonlinearity. For neuron $i$, let $\tilde{n} = \sum_{j \neq i} x_j$ be the spike count of the other units, and define

$$Q(\tilde{n}) = \sum_{j=1}^{N} (-1)^{j+1} C_j \left(\frac{\tilde{n}}{N}\right)^j,$$ (66)

$$\Delta Q(\tilde{n}) = Q(\tilde{n}+1) - Q(\tilde{n}).$$ (67)

The unnormalized joint activities of neurons are

$$p_0 \propto h(\tilde{n}) \exp(-f Q(\tilde{n})) \quad (x_i = 0),$$ (68)

$$p_1 \propto h(\tilde{n}+1) \exp(-f Q(\tilde{n}+1)) \quad (x_i = 1).$$ (69)

We update $x_i$ using the following conditional probability given the state of all other neurons:

$$p(x_i = 1 | \mathbf{x}_{\setminus i}) = \frac{1}{1 + \exp\left(-\log(p_1/p_0)\right)}.$$ (70)

The log-ratio simplifies to

$$
\begin{aligned}
\log \frac{p_1}{p_0} &= \left[\log h(\tilde{n}+1) - \log h(\tilde{n})\right] - f \Delta Q(\tilde{n}) \\
&= \log\left(\frac{\tilde{n}+1}{N-\tilde{n}}\right) - f \Delta Q(\tilde{n}).
\end{aligned}
$$ (71)

One sweep visits all $i = 1, \ldots, N$ in permuted order and applies this update. We obtained 1,000,000 samples.

The resulting spike sequences were then fitted with the state-space kinetic Ising model. Because the data were stationary, we fixed the state noise covariance to zero, $\mathbf{Q}^i = 0$ ($i = 1, \ldots, N$), and omitted hyperparameter optimization. To reduce computation time, the samples were reorganized into $T = 200$ time bins and $L = 5000$ trials, preserving dependencies across consecutive bins within each trial. Under this setting, the fitted state-space model yielded constant parameters across bins. We then generated 500,000 spike sequences by resampling from the fitted model, and compared their population spike-count distribution with that of the original Gibbs-sampled data.

## Data availability

We used the publicly available Allen Brain Observatory: Visual Behavior Neuropixels dataset provided by the Allen Institute for Brain Science: https://portal.brain-map.org/circuits-behavior/visual-behavior-neuropixels. Large precomputed datasets required to reproduce the figures are archived on Zenodo: 10.5281/zenodo.15220108.

## Code availability

The analysis code used in this study is archived on Zenodo and linked to the GitHub repository: https://doi.org/10.5281/zenodo.17504162. For convenient browsing, see the GitHub mirror: https://github.com/KenIshihara-17171ken/Non_equ.

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

## Acknowledgements

The authors thank Shinji Nakaoka for his kind support of this project. This work was supported by JSPS KAKENHI Grant Numbers JP 20K11709, 21H05246, 24K21518, 25K03085 (H.S.).

## Author contributions

K.I. developed the algorithms, implemented the code, and performed the data analyses. H.S. conceived and supervised the project and contributed to analyses. Both authors wrote and revised the manuscript.

## Competing interests

The authors declare no competing interests.
