## [Transparent Peer Review file · Nature Communications]

State-space kinetic Ising model reveals task-dependent entropy flow in sparsely active nonequilibrium neuronal dynamics

Corresponding Author: Professor Hideaki Shimazaki

Version 0:

Reviewer comments:

Reviewer #1

(Remarks to the Author)

The ms by Ishihara and Shimazaki presents a very relevant and necessary novel approach to analyzing non-stationary and nonequilibrium neuronal spiking dynamics data by extending the kinetic Ising model into a state-space framework. In particular, the authors show how entropy flow relates to behavioral states. The study is well-motivated, mathematically rigorous, technical depth and more importantly provides valuable insights into neuronal ensemble dynamics in the mouse visual cortex. I strongly recommend publication of this necessary piece of work. I really enjoyed reading this beautiful extension of the basic stochastic non-equilibrium framework of the authors.

I have only minor comments that the authors could consider:

The parameters of the model are of course not only time-dependent but asymmetric. Would the classical mean-field results be recovered if the parameters would be time-independent and symmetric (so equilibrium). Perhaps, add this as a Suppl. Mat.? It is just an advice, but I'm not requiring to do it.

The validation of the proposed method is primarily done through simulations and application to experimental data. However, it would strengthen the study if the model's performance were compared against existing methods that analyze time-asymmetric neuronal dynamics. For instance, how does it perform relative to steady-state entropy production methods used in previous fMRI or ECoG studies? (or at least to add references to non-equilibrium framework recently published:

[https://www.cell.com/trends/cognitive-sciences/fulltext/S1364-6613\(24\)00075-5](https://www.cell.com/trends/cognitive-sciences/fulltext/S1364-6613(24)00075-5)

<https://journals.aps.org/pre/abstract/10.1103/PhysRevE.108.064410>

<https://www.nature.com/articles/s42003-022-03505-7>

<https://direct.mit.edu/netn/article/7/3/966/114355/The-arrow-of-time-of-brain-signals-in-cognition>

The key result that high-performing mice exhibit increased entropy flow in higher-firing neurons is intriguing. However, it remains unclear whether this pattern is a direct consequence of neural computation related to task performance or a secondary effect of differences in network state. Perhaps discuss potential alternative explanations, such as differential attentional states or task engagement levels?

Excellent work! Congrats

Gustavo Deco

(Remarks on code availability)

Reviewer #3

(Remarks to the Author)

May 7, 2025

Title : State-space kinetic Ising model reveals task-dependent entropy flow in sparsely active nonequilibrium neuronal dynamics

Submitted for publication in Nature Communications

1 Manuscript overview

This manuscript introduces a state-space kinetic Ising model to capture nonstationary and nonequilibrium features in neuronal population activity. The authors propose a mean-field approximation to estimate time-varying entropy flow and apply the method to large-scale neural recordings from mouse visual cortex under active and passive behavioral states. The study is innovative in its methodological development and its application to understanding entropy dissipation dynamics linked to behavioral engagement.

While methodologically innovative and analytically rich, the manuscript currently falls short in conceptual framing and mechanistic insight. It proposes a technically interesting model, but lacks a fully developed hypothesis or critical discussion of generality and its limitations. Strengthening these aspects would significantly enhance its contribution to computational neuroscience and statistical physics of neural systems.

Overall, this article presents an interesting methodological approach; however, its applicability to a broader range of neuroscientific data appears limited. Furthermore, the reported results do not seem particularly compelling due to their high variability. Figures focusing on individual cases, such as Mouse ID: 574078, would be more appropriate in the supplementary material, while results pertaining to the broader population should be emphasized in the main text.

The current state of the manuscript presents important limitations in terms of hypothesis development, mechanistic interpretation, and transparency regarding model limitations. The work needs to address major issues before further consideration, for example the following.

2 Major

• Limited Discussion on the Hypothesis Underpinning the Methodological Approach

The manuscript provides minimal theoretical or empirical justification for why entropy flow (specifically as defined through the kinetic Ising model) should reflect meaningful cognitive or behavioral phenomena in neuronal populations.

There is no formal hypothesis on what the entropy flow dynamics are expected to reveal about the neural computation or behavior prior to analysis. As a result, the interpretation of increased entropy flow in high-performing mice remains post hoc and correlational.

I would suggest to clearly formulate a hypothesis about the expected relationship between entropy flow, spiking sparsity, and cognitive performance, possibly grounded in sparse coding or predictive coding frameworks.

• Limited Discussion of Methodological and Modeling Limitations

The model's coupling parameters are interpreted as proxies for causal interactions, yet the manuscript acknowledges that these do not map onto synaptic or effective connectivity. There is no attempt to systematically analyze or bound the interpretability of these couplings.

While the EM-based inference and mean-field approximations are interesting, the impact of model assumptions (e.g., binning, Gaussian process priors, independence of neurons' parameter dynamics) on the validity of entropy estimates is not rigorously evaluated.

The model is benchmarked only on simulation data that closely matches its own assumptions (e.g., Gaussian sampling of time-varying parameters). There is no analysis of robustness under model misspecification, e.g., when spiking statistics deviate from logistic assumptions or exhibit higher-order correlations.

I suggest to include a more explicit limitations subsection addressing identifiability, model misspecification, and the limits of using statistical constructs to infer neurobiological phenomena.

• Limited Mechanistic Insight into Observed Neural Phenomena

The observed association between increased entropy flow in high-firing neurons and behavioral performance is not mechanistically unpacked. Are these neurons more connected? Do they drive others? Are they receiving stronger input? The current model lacks the granularity to answer these questions.

Similarly, the interpretation of entropy flow as a proxy for "causal" information transfer or "organized structure" lacks specific neurophysiological anchoring. The term "structure" remains vague.

There is no model-based perturbation analysis to assess how altering coupling distributions or firing sparsity would predictably affect entropy flow. This weakens claims about the mechanistic role of sparse coding in modulating nonequilibrium dynamics.

I would suggest to integrate or at least discuss how this framework could be extended to,

multiscale analyses, or mechanistic circuits. Can you comment on other types of data in neuroscience. EEG, MEG, fMRI, LFP's, MEA?

- Better explain the limitations

Physiological Interpretation of Couplings: The first limitation emphasizes that the model's estimated couplings, while useful for capturing directed dynamics, are statistical abstractions and not direct analogs of real synaptic connections. This disconnect is attributed to the simplified nonlinearity in the model (e.g., logistic or tanh functions), which does not fully capture the complex, biophysical behavior of actual neurons. The authors suggest future work could improve biological plausibility—perhaps through more advanced neuron models—without losing the tractability of thermodynamic interpretations. Can you provide a specific example of how this can be done?

Incomplete Use of Thermodynamic Measures: The second limitation concerns the scope of the metrics used. The study focuses on entropy flow as a proxy for time-asymmetry, but this alone does not fully capture the system's dynamical regime. For instance, high entropy flow could reflect different phenomena, such as proximity to criticality or chaos. The authors propose that complementary measures (like conditional entropy) would enrich the interpretive power of their analyses. Can you be more specific about this?

Temporal Scope of the Analysis: The third limitation involves the timescale of the analysis. The model is tested primarily on short timescales, suitable for stimulus-response dynamics, but it remains unclear how well it generalizes to longer-term processes, such as learning or memory consolidation. This suggests a need for future work on scaling the model to capture broader temporal dynamics. How can this be done in practice?

3 Minor

- fig 2 B) and 5B) Estimated coupling parameters..this figures are not understandable, too saturated and it is not cclear the message you want to convey here.
- Fig 3. Estimation error and computational time. these results should be interpreted beyond the technique, they should illuminate a discussion the limitations and broader applications of this method.
- Fig 5A. Is there an active condition? The time points should have a different color.
- Is it possible to assign units to entropy flow?
- In the abstract "violating steady-state assumption." to "...violating the steady-state assumption."
- In the abstract "...directed activity emerged in a fewer neurons..." "...directed activity emerged in fewer neurons..."
- Introduction, line 69: " enabled to reveal details..." to "Thus, the method enabled us to reveal details..."
- Results, Figure 2C caption: "...comparing the true coupling papameters..." to "...comparing the true coupling parameters..."
- Results, line 248: Remove space before comma.
- Discussion, sentence 519: "...statistical constructs and do not represent physiological synaptic connections." to "...statistical constructs that do not necessarily represent physiological synaptic connections."
- "Fig. 5A Left" to "Fig. 5A (left)"

(Remarks on code availability)

Version 1:

Reviewer comments:

Reviewer #1

(Remarks to the Author)

The authors responded properly my comments.
I recommend accepting the ms.

(Remarks on code availability)

Reviewer #3

(Remarks to the Author)

I would like to start by congratulating the authors for the extensive work done to improve the manuscript. The revised version is significantly strengthened: the hypothesis is now clearly articulated, the mechanistic interpretation has been expanded with analytical derivations, model limitations are clearly acknowledged, and additional analyses—including perturbation and cross-image validation—greatly enhance the robustness and generality of the results. I acknowledge the authors for

addressing the majority of my concerns.

That said, I believe there remains one aspect that could benefit from further clarification. While the authors now emphasize the statistical abstraction of couplings and entropy flow, the physiological interpretability of these measures is still only partially grounded. The discussion points toward neuromodulatory influences (e.g., acetylcholine, norepinephrine), which is an excellent step. However, it would further strengthen the manuscript if the authors could more explicitly delineate how future experimental manipulations (e.g., pharmacological perturbations, optogenetic modulation of inhibitory/excitatory balance) might serve to test the predictions derived from entropy flow analyses. This novel insights from this work would help bridge the gap between the current statistical–thermodynamic framework and its experimental validation in neurobiology.

(Remarks on code availability)

Black: Reviewer's comments

Blue: Our responses

Reviewer #1 (Remarks to the Author):

The ms by Ishihara and Shimazaki presents a very relevant and necessary novel approach to analyzing non-stationary and nonequilibrium neuronal spiking dynamics data by extending the kinetic Ising model into a state-space framework. In particular, the authors show how entropy flow relates to behavioral states. The study is well-motivated, mathematically rigorous, technical depth and more importantly provides valuable insights into neuronal ensemble dynamics in the mouse visual cortex. I strongly recommend publication of this necessary piece of work. I really enjoyed reading this beautiful extension of the basic stochastic non-equilibrium framework of the authors.

I have only minor comments that the authors could consider:

The parameters of the model are of course not only time-dependent but asymmetric. Would the classical mean-field results be recovered if the parameters would be time-independent and symmetric (so equilibrium). Perhaps, add this as a Suppl. Mat.? It is just an advice, but I'm not requiring to do it.

We thank the reviewer for raising this important point. Indeed, your insightful comment allowed us to recognize a key limitation of the current mean-field approximation that had not been reported in the initial submission. We have now clarified this limitation in the revised manuscript by adding new Supplementary Information S3 Text and corresponding explanations in the main text and Methods.

When the parameters are time-independent, the model reduces to the steady-state case. Further, if the connections are symmetric, the system is in equilibrium and should exhibit no entropy flow or entropy production.

However, the mean-field method based on the Gaussian approximation employed in this study primarily quantifies the causal strength from mean-field inputs, while remaining insensitive to the detailed structure of the couplings. To illustrate this point, we added a new Supporting Information S3 text, where we derived the steady-state solution of the mean-field method:

$$\sigma_t^{\text{flow}} \approx \sum_i \int \mathcal{D}_z \left(r(g_i + z\sqrt{\Delta_i}) - m_i \right) \cdot z\sqrt{\Delta_i}$$

The first term in the integrand represents deviations of the i-th neuron's activity rate from its long-term average, while the second term represents the fluctuating inputs to the neuron. Thus, the method effectively measures correlations between neuronal responses and input fluctuations. However, this formulation clearly shows that the measure is agnostic to whether the couplings are symmetric or asymmetric, and

depends instead on the magnitude of the couplings. Consequently, it does not yield zero entropy flow for symmetric connections. Intuitively, this mean-field method yields entropy flow under the assumption of fully asymmetric couplings, as is commonly adopted in theoretical studies of the kinetic Sherrington–Kirkpatrick model.

This is an important limitation of applying the current mean-field method. In particular, if differences in entropy flow primarily arise from coupling symmetry versus asymmetry, one must resort to the sampling-based estimation method. We have clearly noted this limitation at the end of the section “*Mean-field method for estimating entropy flow*” in Results as follows:

... The mean-field steady-state solution thus provides an intuitive view of entropy flow as a measure of the neuron’s causal responsiveness to input fluctuations. However, this equation clarifies that the approximation depends mainly on the magnitudes of the field and coupling parameters and remains insensitive to the detailed coupling structure. It should therefore be applied with caution when asymmetry is the primary source of entropy flow.

However, it should also be noted that, in our analysis, the strength of coupling asymmetry, relative to the baseline established by shuffled data, did not differ significantly between the active and passive conditions (Fig. 7I; this part has been revised for clarity). The contribution to the entropy flow in the dataset arises instead from changes in coupling variability, which justifies using the present mean-field method in this context. To clarify this point, we have added the corresponding explanation in the revised manuscript:

The lack of change in asymmetry in the effective couplings validates the use of the proposed mean-field method for comparing the coupling effect, which primarily arises from variability changes rather than asymmetry.

Finally, we added the improvement of the mean-field method as one of the major future challenges in the Discussion:

Future improvements through parallelization, optimized algorithms, and refined mean-field approaches could extend its applicability and enhance entropy flow estimation.

We thank the reviewer again for pointing out this subtle but essential limitation of the current mean-field method, which opens an avenue for developing improved mean-field approaches in future work.

The validation of the proposed method is primarily done through simulations and application to experimental data. However, it would strengthen the study if the model’s performance were

compared against existing methods that analyze time-asymmetric neuronal dynamics. For instance, how does it perform relative to steady-state entropy production methods used in previous fMRI or ECoG studies? (or at least to add references to non-equilibrium framework recently published:

Kringelbach, M. L., Perl, Y. S., & Deco, G. (2024). The thermodynamics of mind. *Trends in Cognitive Sciences*, 28(6), 568-581.

Deco, G., Lynn, C. W., Sanz Perl, Y., & Kringelbach, M. L. (2023). Violations of the fluctuation-dissipation theorem reveal distinct nonequilibrium dynamics of brain states. *Physical Review E*, 108(6), 064410.

Deco, G., Sanz Perl, Y., Bocaccio, H., Tagliazucchi, E., & Kringelbach, M. L. (2022). The INSIDEOUT framework provides precise signatures of the balance of intrinsic and extrinsic dynamics in brain states. *Communications Biology*, 5(1), 572.

Deco, G., Sanz Perl, Y., de la Fuente, L., Sitt, J. D., Yeo, B. T., Tagliazucchi, E., & Kringelbach, M. L. (2023). The arrow of time of brain signals in cognition: Potential intriguing role of parts of the default mode network. *Network Neuroscience*, 7(3), 966-998.

Thank you for the thoughtful suggestion. In principle, our state-space kinetic Ising framework could be applied to fMRI or ECoG by appropriate binarization of the signals. A fair, head-to-head benchmark against macroscopic, steady-state irreversibility metrics (e.g., arrow-of-time/TENET or FDT-based approaches) is non-trivial, and goes beyond the scope of this manuscript. In the revised manuscript, we added the following paragraph on the possible application of our method to EEG, fMRI, and ECoG in Discussion that included and highlighted the citations above:

EEG, fMRI, and ECoG studies suggest that steady-state entropy production and related irreversibility metrics covary with consciousness level and cognitive load, and they reveal large-scale directed temporal structure `\cite{perl2021nonequilibrium, deco2023violations, deco2022insideout, deco2023arrow, sekizawa2024decomposing, kringelbach2024thermodynamics}`. For example, in human fMRI, violations of the fluctuation–dissipation theorem are larger during wakefulness than deep sleep, and larger during tasks than rest `\cite{deco2023violations}`. Arrow-of-time analyses likewise show stronger temporal asymmetry during tasks than rest and identify a cortical hierarchy of asymmetry `\cite{deco2023arrow}`. Our state-space kinetic Ising model complements these steady-state, macroscopic approaches by estimating entropy flow directly from spiking data without assuming stationarity, potentially illuminating the lower-level mechanisms of mesoscopic/microscopic circuit dynamics. In parallel, Ising and energy-landscape methods have been successfully applied to neuroimaging and electrophysiological data to characterize correlation structure and attractor basins of large-scale brain networks `\cite{watanabe2013pairwise, ezaki2017energy, masuda2025energy, watanabe2025noninvasive}`. By contrast, our framework explicitly quantifies

time-asymmetric entropy flow in non-stationary spiking populations, thereby complementing energy-landscape analyses of macroscopic stability with directed causal dynamics. In principle, the same approach could be extended upward in scale to local field potentials (LFPs), multi-electrode arrays (MEAs), or coarse-grained EEG/ECOG recordings, enabling multiscale integration of nonequilibrium dynamics from circuit to whole-brain levels.

The key result that high-performing mice exhibit increased entropy flow in higher-firing neurons is intriguing. However, it remains unclear whether this pattern is a direct consequence of neural computation related to task performance or a secondary effect of differences in network state. Perhaps discuss potential alternative explanations, such as differential attentional states or task engagement levels?

We thank the reviewer for pointing out the gap between neural computation and network states. Your insightful comment (and suggestions by Reviewer 3) motivated us to conduct additional analyses more directly connected to sparse coding, while expanding the data sets to include neuronal responses to more images. Although the predictive ability of entropy flow with respect to behavioral performance of individual mice remains, these extended analyses revealed new findings that required us to revise our view on the relative contribution of the low and firing rate neurons to entropy flow and its relation to behavioral performance. Accordingly, we have revised the relevant section of the manuscript and clarified the conclusions.

Fig 10. Correlations between neural dynamics and behavioral performance. **A** histogram of firing-rate differences (active–passive) across neurons for mouse 574978 with an image im036r. A skewness of this distribution was used as a sparsification index. **B** The skewness vs behavioral performance as measured by d-prime for 37 mice exposed to three images.

First, we asked whether the active and passive conditions could be inferred from firing rates alone (Fig.10A,B). We found that the sparsity measure of firing-rate distributions was strongly correlated with behavioral performance (d'). This analysis included responses of 37 mice to two additional images. The results provide more direct evidence that animals employing sparser representations of images exhibit higher performance, even without reference to entropy flow. Previously, we had reported that entropy flow predicted task performance, but these new results show that performance can be predicted from shifts in firing-rate distributions alone.

Next, we re-examined the predictive performance of entropy flow, which reflects both firing rates and couplings. By analyzing the composition of total entropy flow, we found that neurons receiving high mean input tended to show reduced input variability, whereas neurons with low mean input displayed increased variability (Fig.10C, D). Thus, total entropy flow arises from two separable sources: (i) mean-input-driven contributions associated with high firing, and (ii) variability-driven contributions that can be substantial even in low-firing neurons.

To focus on the latter, we normalized entropy flow by activity rate. This normalization reduced the direct dependence on mean firing and highlighted variability-driven contributions. We found that shift in mean entropy flow per activity rate (active - shuffle) was significantly correlated with behavioral performance (Fig.10E), using neuronal responses from 37 mice across three images. Importantly, this correlation disappeared in trial-shuffled data (Fig.10F), confirming that couplings contributed beyond activity-rate sparsity. This result suggests that the thermodynamic cost per spike, shaped in part by couplings, is related to behavioral performance.

Fig 10. Correlations between neural dynamics and behavioral performance. **E** Entropy flow normalized by activity rate vs behavioral performance. **F** Entropy flow normalized by activity rate obtained from trial-shuffled data vs behavioral performance. In E and F, there is one outlier mouse below -20, which was included in the statistical analysis.

We also revisited the alternative explanation that behavioral performance could be linked primarily to changes in high-firing neurons. While our earlier analysis (one image) showed such a relationship, the effect was not consistent across the additional images. Thus, high-firing-based changes alone cannot explain performance differences. Instead, the more robust association with entropy flow per activity rate supports a complementary role of variability-driven, coupling-mediated contributions—particularly in low-firing neurons—in explaining behavioral performance.

In summary, we made the following changes to the manuscript:

- We revised the section on entropy flow and behavioral performance.
- We moved the earlier analysis to Supporting Information S5 Text.

Finally, the analyses above remain correlational and cannot determine whether the observed pattern reflects a direct neural computation related to task performance or a secondary effect of network state, as the reviewer noted. Following the reviewer's suggestion, we added the following remark at the end of the Discussion, together with possible tests to address the question.

Future work should determine whether this pattern reflects a direct computational mechanism or a secondary consequence of network state (e.g., attention or arousal). Importantly, the proposed method further yields testable predictions for information coding. For instance, if entropy flow per spike indeed relates to computation, then (i) neurons whose receptive fields match the presented image features should show selectively higher entropy flow per spike, or (ii) population decoding accuracy should improve when restricting analysis to neurons with higher entropy flow per spike. These predictions provide avenues to experimentally validate the computational role of entropy flow.

We sincerely appreciate the reviewer's insight, which prompted us to refine our conclusion that had previously been based on only one image.

Excellent work! Congrats
Gustavo Deco

Thank you for highlighting the critical points and for encouraging comments.

Reviewer #3 (Remarks to the Author):

Referee Report, manuscript NCOMMS-25-20836

May 7, 2025

Title : State-space kinetic Ising model reveals task-dependent entropy flow in sparsely active nonequilibrium neuronal dynamics

Submitted for publication in Nature Communications

1 Manuscript overview

This manuscript introduces a state-space kinetic Ising model to capture nonstationary and nonequilibrium features in neuronal population activity. The authors propose a mean-field approximation to estimate time-varying entropy flow and apply the method to large-scale neural recordings from mouse visual cortex under active and passive behavioral states. The study is innovative in its methodological development and its application to understanding entropy dissipation dynamics linked to behavioral engagement.

While methodologically innovative and analytically rich, the manuscript currently falls short in **conceptual framing and mechanistic insight**. It proposes a technically interesting model, but lacks a fully developed **hypothesis or critical discussion of generality and its limitations**. Strengthening these aspects would significantly enhance its contribution to computational neuroscience and statistical physics of neural systems.

Overall, this article presents an interesting methodological approach; however, its applicability to **a broader range of neuroscientific data** appears limited. Furthermore, the reported results **do not seem particularly compelling due to their high variability**. Figures focusing on individual cases, such as Mouse ID: 574078, would be more appropriate in the supplementary material, while results pertaining to the broader population should be emphasized in the main text.

The current state of the manuscript presents important limitations in terms of **hypothesis development, mechanistic interpretation, and transparency regarding model limitations**. The work needs to address major issues before further consideration, for example the following.

We thank the Reviewer for the constructive and helpful comments on our manuscript. In the revised version, we have strengthened the conceptual framing and mechanistic interpretation, and clarified the limitations, reliability, and broader applicability of our approach. Specifically:

Hypothesis and framing – We now clearly articulate a hypothesis in the *Introduction*: “we expected that high-performing animals would exhibit greater entropy flow per spike,

consistent with efficient coding.” In the *Discussion*, we also provide testable predictions on information coding.

Mechanistic interpretation – We added an analytical solution of steady-state entropy flow for mechanistic interpretation (Supporting Information **S3 Text**) and a data-driven analysis using this measure (*Results*; Fig. 10C,D), showing how entropy flow arises from mean inputs and their fluctuations. In the *Discussion*, we further relate our findings to reported effects of neuromodulators.

Model limitations – We added a subsection on model misspecification (*Results*; Fig. 5) showing that the kinetic Ising model fails to capture heavy-tailed statistics due to its assumptions of pairwise interactions and synchronous updates. In the *Discussion*, we added a paragraph highlighting the need for asynchronous and higher-order models.

Reliability of results – We expanded the analysis of entropy flow and behavioral performance to include additional image sets (*Results*; Fig. 10) and introduced a model-based perturbation analysis (*Results*; Fig. 9). Throughout, conclusions in each subsection are based on population-level analyses and comparisons with shuffled-subtracted data. The single-mouse analysis (Fig. 6) was retained only to illustrate the procedure and is now clearly separated as its own subsection.

Broader applicability – We introduced new discussion on how our framework complements recent EEG, fMRI, and ECoG studies that use irreversibility metrics or Ising-model energy landscapes. We also clarified that the state-space kinetic Ising model can be applied to longer timescales, such as learning paradigms.

Our detailed point-by-point responses are provided below.

2 Major

- Limited Discussion on the Hypothesis Underpinning the Methodological Approach

The manuscript provides minimal theoretical or empirical justification for why entropy flow (specifically as defined through the kinetic Ising model) should reflect meaningful cognitive or behavioral phenomena in neuronal populations. There is no formal hypothesis on what the entropy flow dynamics are expected to reveal about the neural computation or behavior prior to analysis. As a result, the interpretation of increased entropy flow in high-performing mice remains post hoc and correlational.

I would suggest to clearly formulate a hypothesis about the expected relationship between entropy flow, spiking sparsity, and cognitive performance, possibly grounded in sparse coding or predictive coding frameworks.

We thank the reviewer for highlighting the need to make our hypothesis explicit. In the revised manuscript, we now clearly articulate the following hypothesis in Introduction:

Given this method, we hypothesized that entropy flow, as estimated with the kinetic Ising framework, reflects the capacity of neural populations to perform meaningful computation under energetic and behavioral-time constraints \cite{wolpert2024stochastic}. Specifically, we expected that high-performing animals would exhibit greater entropy flow per spike, consistent with efficient coding.

This prediction was grounded by two additional analyses:

Theoretical analysis (new): We derived an analytic expression for steady-state entropy flow in the kinetic Ising system, which quantifies correlations between neural responses and input fluctuations (Please see also our response to 'Limited Mechanistic Insight into ...'). This formulation highlights entropy flow as a measure of causal responsiveness at the single-neuron level—a quantity that provides the correlation underlying Hebbian plasticity and, thus, learning performance. This analysis is presented in Supporting Information S3 Text. We also introduced this equation (Eq. 20) after we introduced the mean-field entropy flow in Results, where we stated:

The mean-field steady-state solution thus provides an intuitive view of entropy flow as a measure of the neuron's causal responsiveness to input fluctuations -- a quantity that captures the correlation underlying Hebbian plasticity in neural systems.

Further theoretical support comes from the thermodynamic uncertainty relation and the thermodynamic speed-limit theorems, which show that entropy production sets fundamental bounds on both the precision of stochastic dynamics and the rate at which probability distributions can evolve—implying that precise computation within finite time necessarily requires dissipation. To make these points clear, we added the following sentence to the Introduction:

Further, the thermodynamic uncertainty relation \cite{barato2015thermodynamic, gingrich2016dissipation, proesmans2017discrete, aguilera2025inferring} and the thermodynamic speed-limit theorem \cite{shiraishi2018speed, van2023thermodynamic} show that dissipation sets fundamental bounds on how precisely and rapidly systems can evolve.

Experimental analysis (new): We performed new analyses by extending the data sets to include neural responses to two other images. We used the normalized entropy flow per spike as a measure to correlate with individual mice performance, which is more consistent with efficient coding. Consistent with the hypothesis stated above, we observed that the normalized entropy flow per spike was greater in mice with higher task

performance. **Please also see our response to the comment 'The key result that ...' by Reviewer 1.**

We note that, when we extended our previous analysis on the relation between entropy flow associated with higher-firing neurons and behavior by including additional images, we found that the relation was not supported (not significant in the other two data sets). Thus, while reporting this result (Supporting Information S3 Text), we present a revised analysis that entropy flow *per spike* is related to behavioral performance.

Furthermore, the framework generates testable predictions on sparse coding. To make this point clear, we added the following sentences in the Discussion.

The proposed method further yields testable predictions on information coding. For instance, if entropy flow per spike indeed relates to computation, then (i) neurons whose receptive fields match the presented image features should show selectively higher entropy flow per spike, or (ii) population decoding accuracy should improve when restricting analysis to neurons with higher entropy flow per spike. These predictions provide avenues to experimentally validate the computational role of entropy flow.

- Limited Discussion of Methodological and Modeling Limitations

The model's coupling parameters are interpreted as proxies for causal interactions, yet the manuscript acknowledges that these do not map onto synaptic or effective connectivity. There is no attempt to systematically analyze or bound the interpretability of these couplings.

While the EM-based inference and mean-field approximations are interesting, the impact of model assumptions (e.g., binning, Gaussian process priors, independence of neurons' parameter dynamics) **on the validity of entropy estimates is not rigorously evaluated.**

The model is benchmarked only on simulation data that closely matches its own assumptions (e.g., Gaussian sampling of time-varying parameters). There is no analysis of robustness under model misspecification, **e.g., when spiking statistics deviate from logistic assumptions or exhibit higher-order correlations.**

I suggest to include a more explicit limitations subsection addressing **identifiability, model misspecification**, and **the limits of using statistical constructs to infer neurobiological phenomena.**

We thank the reviewer for clearly pointing out this limitation. In response to the reviewer's suggestion, we have added a dedicated subsection on *Model limitations* in Results. In this section, we conducted additional analyses on model misspecification, examining the consequences of spiking statistics that deviate from logistic assumptions, exhibit higher-order correlations, or violate the assumption of conditional independence

across neurons. These issues also bear directly on identifiability and highlight the limits of using statistical constructs to infer neurobiological phenomena, which we have addressed in the revised Discussion.

Prior studies have already examined the consequences of fitting kinetic Ising models to networks of leaky integrate-and-fire neurons, showing systematic discrepancies between inferred couplings and true connectivity [1,2]. Building on recent theoretical developments [3], we now turn to an alternative population model based on exponential-family distributions, which incorporate higher-order interactions in a mathematically transparent manner. The model was designed so that the distribution of the population spike-count histogram exhibits sparse and widespread features as reported in the literature (Fig.5A, green). This model and following sampling method was introduced in the new subsection in Method.

We performed Gibbs sampling from this distribution (Fig.5A, blue circle) whose dynamics map onto a recurrent network in which neurons follow a supralinear activation function (Fig.5B, green). We then fitted the state-space kinetic Ising model to this activity (Fig.5A, yellow). The result shows that the kinetic Ising model fails to reproduce the observed spike-count histogram for two main reasons. First, the model is restricted to pairwise interactions (i.e., logistic activation function with linear synaptic summation, Fig.5B, yellow) and thus cannot capture higher-order dependencies. It is known that reproducing widespread spike-count histograms in large populations requires interactions of all orders [4]. By contrast, the pairwise model tends to concentrate probability mass at only a few points, often overestimating the tail of the distribution, primarily because it neglects the negative triplewise interactions observed in both the model and real neurons [3].

Fig 5. Analysis of the model miss-specification. **A** The population spike count histogram of $N = 30$ neurons following the shifted-geometric model with a sparseness parameter $f = 20$ and $\tau = 0.8$ (empirical distribution obtained by the Gibbs sampling in green; theoretical probabilities in black). The red line represents a distribution obtained from the state-space kinetic Ising model fitted to the Gibbs sampling data. **B** The activation function of the shifted-geometric model with $f = 20$ and $\tau = 0.8$ (green) and that of the kinetic Ising model using the average of the fitted field and coupling parameters.

Second, the conditional independence of the kinetic Ising model enforces synchronous update rule across neurons within each step. By contrast, Gibbs sampling with sequential (or randomly ordered) updates guarantees detailed balance and allows neurons to incorporate the most recent changes, enabling activity to propagate within a sweep and generate synchronous states. Because the kinetic Ising model updates all neurons simultaneously from the previous state, it lacks this recruitment mechanism and consequently fails to drive synchronous activity appropriately.

We have added this analysis as a subsection at the end of the simulation analysis in the Results section, framed with the following opening and closing statements to clearly highlight the model's limitations:

We end simulation analysis by acknowledging that assumptions of the kinetic Ising framework, in particular pairwise couplings and conditional independence, represent simplifications that may not faithfully capture neural population statistics.

.....

The results highlight that caution is warranted in applying the kinetic Ising framework: although it offers a tractable statistical description, its simplifying assumptions constrain the neural dynamics it can represent. In particular, entropy flow estimates should be regarded as quantities defined under the pairwise and synchronous-update assumptions, rather than unbiased measures of entropy flow in real populations.

These mechanisms have direct implications for the effect of bin-size selection, therefore identifiability of the model. As bins grow larger, within-bin cascades (if exists) are merged into single counts, making recruitment-driven co-activations appear as heavy-tailed spike-count histograms—particularly when neuronal activation functions include a supralinear component. By contrast, the kinetic Ising model, with its synchronous updates and enforced conditional independence within each step, cannot capture such intra-bin propagation. As a result, increasing bin size leads it to systematically fail to estimate the probability of large- n events, often failing to model the heavy tails. Choosing smaller bin sizes alleviates this problem by preserving sequential recruitment, but then may fail to capture slower, history-dependent interactions or coordinated activity that unfolds over longer timescales. Thus, bin-size selection for fitting the Markov model entails a fundamental trade-off between resolving fine-scale cascades and representing broader network-level dependencies.

These considerations suggest that future models should be designed to permit asynchronous updates. The generalized linear models (GLMs) and related point-process frameworks provide a natural asynchronous alternative, since spikes are modeled in continuous time (or with sufficiently fine-grained bins) and each event influences others through coupling kernels. These models therefore avoid the synchrony limitations of the

discrete-time kinetic Ising formulation. Developing practical estimators of entropy flow for such models remains an important challenge for future work.

Accordingly, we revised the paragraph in Discussion on identifiability, model misspecification, and the limits of using statistical constructs as follows:

The kinetic Ising-based framework should also be viewed in light of its limitations. While analytically tractable, it imposes strong assumptions, pairwise couplings and conditional independence, that simplify neural dynamics but restrict interpretability. Our model-misspecification analysis (Fig.~\ref{fig:model_perturbation_analysis}) showed that reproducing the heavy-tailed spike-count statistics observed in real populations requires higher-order interactions; neglecting these leads to systematic biases, particularly in the tails. Likewise, synchronous updates imposed by the conditional independence obscure cascade-like recruitment within bins, producing bin-size-dependent distortions: large bins exaggerate heavy tails by merging cascades, while small bins preserve cascades but miss slower interactions distant in time. Thus, bin size entails a fundamental trade-off between resolving fine-scale recruitment and capturing broader dependencies.

These limitations motivate extensions beyond the synchronous pairwise framework. The generalized linear models (GLMs) and related point-process models provide a natural asynchronous alternative, since spikes are modeled in fine-grained bins or continuous time and influence others through coupling kernels, but entropy flow in such history-dependent systems requires full path probabilities, making estimation challenging. More broadly, fitted couplings and entropy flow should be regarded as statistical summaries of nonequilibrium dynamics, not direct measures of synaptic connectivity or mechanism. Future work must relax these constraints --by permitting asynchronous updates, incorporating higher-order dependencies, and developing principled estimators of entropy flow in non-Markovian settings --while remaining clear about the limits of inference when bridging statistical abstractions with physiology. For example, the alternating-shrinking higher-order interaction model (Eq.~\ref{eq:alternating_shrinking_hoi_model}) could be extended to include asymmetric couplings, potentially with asynchronous updates.

We thank the reviewer again for this valuable suggestion, which allowed us to clarify the scope and limitations of the kinetic Ising framework and to outline concrete directions for future extensions.

[1] Roudi, Y., Tyrcha, J., & Hertz, J. (2009). Ising model for neural data: model quality and approximate methods for extracting functional connectivity. *Physical Review E—Statistical, Nonlinear, and Soft Matter Physics*, 79(5), 051915.

[2] Cocco, S., Leibler, S., & Monasson, R. (2009). Neuronal couplings between retinal ganglion cells inferred by efficient inverse statistical physics methods. *Proceedings of the National Academy of Sciences*, 106(33), 14058-14062.

[3] Rodriguez-Dominguez, U., & Shimazaki, H. (2025). Modeling higher-order interactions in sparse and heavy-tailed neural population activity. *Neural Computation in press* (arXiv:2308.13257)

[4] Amari, S. I., Nakahara, H., Wu, S., & Sakai, Y. (2003). Synchronous firing and higher-order interactions in neuron pool. *Neural computation*, 15(1), 127-142.

- Limited Mechanistic Insight into Observed Neural Phenomena

The observed association between increased entropy flow in high-firing neurons and behavioral performance is not mechanistically unpacked. Are these **neurons more connected**? Do they drive others? Are they receiving stronger input? The current model lacks the granularity to answer these questions.

We agree that the previous manuscript lacked sufficient mechanistic insight into the fitted parameters and their relation to increased entropy flow. In the revised manuscript, we explicitly provide an analytical solution of the steady-state mean-field entropy flow, together with a solution quantifying the contribution of firing rate changes (Supporting Information S3 Text). The steady-state mean-field entropy flow can be written as

$$\sigma_t^{\text{flow}} \approx \sum_i \Delta_i \left(\int \mathcal{D}_z r'(g_i + z\sqrt{\Delta_i}) \right)$$

where $r'(h) = r(h)(1 - r(h))$. It computes the correlation of a neuron's response to a fluctuating input, which is a function of the mean (current) input g_i and its fluctuation Δ_i :

$$g_i = \theta_i + \sum_j \theta_{ij} m_j,$$

$$\Delta_i = \sum_j \theta_{ij}^2 m_j (1 - m_j).$$

Under the low firing rate regime ($m_j < 0.5$), both the mean and the fluctuations of the input current increase entropy flow.

Fig 10. Correlations between neural dynamics and behavioral performance. **C** Mean-field rate as a function of time-average mean g_i and variability Δ_i of their inputs for a mouse 574978. The color map shows theoretical mean-field rate under the steady-state (Eq. 19). **D** Mean-field entropy flow of individual neurons as a function of time-average mean and variability of their inputs. The color map shows theoretical entropy flow under the steady-state (Eq. 64).

We considered that showing the composition of entropy flow for neurons with high mean input and/or high input fluctuations would provide a clearer mechanistic picture underlying entropy flow, although here “mechanistic” should be interpreted under the model’s assumptions. In the revised manuscript, we therefore present the entropy flow of individual neurons as a function of g_i and Δ_i , overlaid on the contour map obtained from the analytical solution. This theoretical map clearly shows that increased variability is necessary to maintain entropy flow under reduced mean input.

From the fitted parameters, we found that low mean input to a neuron tends to be associated with increased variability of its inputs. In contrast, neurons with high mean input (which nearly corresponds to high-firing neurons; see Fig. 10C) received less variable inputs. These results suggest that total entropy flow is shaped not only by high-input neurons but also by low-input neurons with high variability, which we stated as:

These patterns imply two separable sources of entropy flow: (i) mean-input–driven contributions that track high firing, and (ii) variability-driven contributions that can be substantial even at low firing. To more focus on the latter, we considered entropy flow per activity rate. This normalization reduces the direct dependence on mean rate and makes variability-driven effects, particularly those arising in low-rate neurons, observable on equal footing with high-rate effects.

This observation prompted us to re-examine the contribution of individual neurons to entropy flow and its relation to behavioral performance, using normalized entropy flow per spike, while carefully addressing statistical noise in coupling estimation.

We are grateful to the reviewer for this insightful comment, which prompted us to add new analyses and theoretical clarification, thereby providing a clearer mechanistic account of how input mean and variability jointly shape entropy flow and its link to behavioral performance.

Similarly, the interpretation of entropy flow as a proxy for “causal” information transfer or “organized structure” lacks specific neurophysiological anchoring. The term “structure” remains vague.

We agree that the notion of an “organized structure” lacked direct neurophysiological anchoring. The increase in entropy flow can arise either from **strong or variable couplings** among neurons, or from temporal reductions in firing rates, as is evident from the analytical expression of entropy flow (Eq. 11). The impact of heterogeneous couplings can also be confirmed from the exact solution of the fully asymmetric SK model reported previously [1].

Our analyses showed that the coupling-related increase in entropy flow during the active condition is primarily attributable to increased variability of the couplings in sparse population activity (Fig.7H). How such heterogeneous *functional* couplings relate to concrete physiological mechanisms remains to be elucidated. We speculate that regulation of asynchronous activity in an excitatory–inhibitory balanced network, potentially mediated by neuromodulators such as acetylcholine or noradrenaline, could underlie the transition between the active and passive states. In Discussion, we have already provided prior reports of increased heterogeneous activities during the active condition and its correlation to behavioral performance (Montijn et al. 2015). Next to this paragraph, we added the following paragraph to provide a neurophysiological anchoring of the observed sparser activity and coupling-induced temporal asymmetry during active engagement, focusing on neuromodulators:

The observed shift in cortical activity largely aligns with the effects of neuromodulators, such as acetylcholine (ACh) \cite{runfeldt2014acetylcholine, chen2015acetylcholine} and norepinephrine (NE) \cite{reitman2023norepinephrine}, that alter local circuit interactions and global activity patterns, thereby regulating transitions such as quiet–active, and inattentive–attentive states \cite{lee2012neuromodulation, mccormick2020neuromodulation}. For example, Runfeldt et al. \cite{runfeldt2014acetylcholine} demonstrated that spontaneous network events became more sparse under ACh, as the probability of individual neurons participating in circuit activity was markedly reduced. In addition, ACh altered the temporal recruitment of neurons, delaying their activation relative to thalamic input and prolonging the window during which stereotyped activity propagated through local circuits. These findings indicate that ACh reorganizes cortical circuits into sparser and temporally extended modes of activity, which may

underlie the sparser population activity and enhanced temporal asymmetry that we observed during task engagement. However, we did not observe the previously reported decoupling of neuronal activity (Fig.~\ref{supp_fig:mean_coupling_BEH}), which may suggest the involvement of additional mechanisms beyond those described above.

Most previous studies reported decorrelation (greater independence) of neuronal activity under ACh application. Since we had not reported the mean values of couplings between active and passive conditions, we added this as a new Supporting Information S5 Fig. Interestingly, we found a slight but significant increase in the mean couplings across all mice, which might indicate the involvement of additional mechanisms as pointed out above.

We also reviewed the manuscript to ensure that the terms “organized structure” and “causal information transfer” were not used as proxies for entropy flow. We do not refer to entropy flow as a proxy for causal information transfer. To make “structured” unambiguous, we revised the theoretical section “Estimating entropy flow” to read “structured (i.e., lower-entropy) states,” clarifying that “structured” denotes reduced entropy:

This indicates that the positive total entropy flow enables the systems to be more structured (i.e., lower entropy) at the final time step than at the initial time step.

We thank the reviewer again for prompting us to articulate clearer neurophysiological underpinnings of the coupling-related increase in entropy flow, which led us to strengthen the experimental context of our manuscript.

[1] Aguilera, M., Igarashi, M., & Shimazaki, H. (2023). Nonequilibrium thermodynamics of the asymmetric Sherrington-Kirkpatrick model. *Nature Communications*, 14(1), 3685.

There is no **model-based** perturbation analysis to assess how altering coupling distributions or firing sparsity would predictably affect entropy flow. This weakens claims about the mechanistic role of sparse coding in modulating nonequilibrium dynamics.

We thank the reviewer for this valuable suggestion. In the revised manuscript, we added a new figure (Fig.9) and a corresponding paragraph describing a model-based perturbation analysis.

We consider two complementary approaches to such perturbation analyses. The first, based on theory or simulation, involves systematically changing model parameters and examining how these changes affect entropy flow. One of the author previously pursued

this direction by deriving and analyzing the exact formula of entropy flow in the Sherrington–Kirkpatrick (SK) model using path-integral and saddle-node methods [1].

The second approach perturbs models fitted directly to empirical data. Specifically, by scaling the field and coupling parameters estimated from real data, one can assess whether the observed activity maximizes particular statistical properties. This approach is commonly used to test for criticality in neuronal dynamics, for example by computing the heat capacity [2]. Given our prior theoretical work, we considered the perturbation analysis of the fitted model to be the more informative option here. This analysis introduced as a new subsection 'Model-based perturbation analysis' in Results yielded the following notable results (Fig.9).

First, the entropy flows during stimulus presentation and the waiting period (gray image) exhibited significantly different behaviors under perturbation. The increase in entropy flow induced by firing rate reduction during stimulus presentation persisted as the scaling parameter β increased. This effect arises from differences in the sensitivity of two consecutive activity rates to β (Eq.178 in Supporting Information S3 Text). By contrast, during the waiting period we observed increased entropy flow only at $\beta < 1$, while the stimulus presentation period remained relatively stationary.

Fig 9. Model-based perturbation analysis **A** Entropy flow σ_t of mouse 574078 in the active condition, computed after rescaling the fitted parameters as $\hat{\theta} \rightarrow \beta \hat{\theta}$. The dashed line indicates $\beta = 1$. **B** Forward entropy flow, $\sigma_t^{\text{forward}}$. **C** Reverse entropy flow, $\sigma_t^{\text{backward}}$. **D** Shuffle-subtracted entropy flow in the active condition, isolating the contribution of interactions beyond firing rate dynamics. **E** Shuffle-subtracted entropy flow in the passive condition, showing interaction-driven peaks at $\beta < 1$. **F** Comparison of shuffle-subtracted entropy flow between active and passive conditions across all mice, showing significantly higher values in the active condition (Wilcoxon test, $p = 1.455 \times 10^{-10}$). Shuffle-subtracted entropy flow is obtained over a low-gain range $\beta \in [0.2, 1.0]$ and across all bins. Lines connect active (left) to passive (right) for each mouse.

By subtracting the entropy flow calculated from trial-shuffled data, which preserve only firing rate dynamics, we confirmed that the increase in entropy flow during stimulus presentation was attributable to firing rate changes, whereas the increase at $\beta < 1$ arose from interactions, since the former disappeared while the latter remained after shuffle subtraction. The interaction-driven entropy flow during the waiting period was maximized at $\beta < 1$, while both forward and reverse entropy decreased as β increased. These results indicate that this subsampled neural population is not in a quasi-deterministic chaotic regime but rather in a subcritical regime.

More interestingly, the interaction-driven entropy flow revealed by parameter scaling differed significantly between the active and passive conditions (Fig.9). Notably, our previous analysis at $\beta = 1$ showed no difference in shuffle-subtracted (i.e., interaction-driven) entropy flow between the two states (Fig.8C). However, examining values of $\beta \neq 1$ uncovered clear and significant differences.

We sincerely thank the reviewer for this insightful comment, which guided us in uncovering previously hidden statistical differences in the fitted models.

[1] Aguilera, M., Igarashi, M., & Shimazaki, H. (2023). Nonequilibrium thermodynamics of the asymmetric Sherrington-Kirkpatrick model. *Nature Communications*, 14(1), 3685.

[2] Tkačik, G., Mora, T., Marre, O., Amodei, D., Palmer, S. E., Berry, M. J., & Bialek, W. (2015). Thermodynamics and signatures of criticality in a network of neurons. *Proceedings of the National Academy of Sciences*, 112(37), 11508-11513.

[3] Aguilera, M., Moosavi, S. A., & Shimazaki, H. (2021). A unifying framework for mean-field theories of asymmetric kinetic Ising systems. *Nature communications*, 12(1), 1197.

I would suggest to integrate or at least discuss how this framework could be extended to, multiscale analyses, or mechanistic circuits. Can you comment on other types of data in neuroscience. EEG, MEG, fMRI, LFP's, MEA?

We have added the following paragraph discussing how the proposed framework could, in principle, be extended to multiscale analyses, drawing on prior applications of Ising and related approaches to other data modalities in neuroscience.

EEG, fMRI, and ECoG studies suggest that steady-state entropy production and related irreversibility metrics covary with consciousness level and cognitive load, and they reveal large-scale directed temporal structure
\cite{perl2021nonequilibrium, deco2022insideout, delafuente2022temporal, deco2023violations, gilson2023entropy, deco2023arrow,

sekizawa2024decomposing, kringelbach2024thermodynamics}. For example, in human fMRI, violations of the fluctuation–dissipation theorem are larger during wakefulness than deep sleep, and larger during tasks than rest \cite{deco2023violations}. Arrow-of-time analyses likewise show stronger temporal asymmetry during tasks than rest and identify a cortical hierarchy of asymmetry \cite{deco2023arrow}. Our state-space kinetic Ising model complements these steady-state, macroscopic approaches by estimating entropy flow directly from spiking data without assuming stationarity, potentially illuminating the lower-level mechanisms of mesoscopic/microscopic circuit dynamics. In parallel, equilibrium Ising and energy-landscape methods have been successfully applied to binarized neuroimaging and electrophysiological signals to characterize correlation structure and attractor basins of large-scale brain networks \cite{watanabe2013pairwise, ezaki2017energy, masuda2025energy, watanabe2025noninvasive}. By contrast, our framework explicitly quantifies time-asymmetric entropy flow in non-stationary binary signals, complementing energy-landscape analyses of macroscopic stability with measures of time-dependent causal dynamics. In principle, the same approach could be extended upward in scale to local field potentials (LFPs), multi-electrode arrays (MEAs), or coarse-grained EEG/ECOG recordings, enabling multiscale analysis of nonequilibrium dynamics from circuit to whole-brain levels. However, the state-space method still faces limits in computational time and scale, constraining its use for these large-scale signals. Future improvements through parallelization, optimized algorithms, and refined mean-field approaches could extend its applicability and enhance entropy flow estimation.

- Better explain the limitations

Physiological Interpretation of Couplings: The first limitation emphasizes that the model’s estimated couplings, while useful for capturing directed dynamics, are statistical abstractions and not direct analogs of real synaptic connections. This disconnect is attributed to the simplified nonlinearity in the model (e.g., logistic or tanh functions), which does not fully capture the complex, biophysical behavior of actual neurons. The authors suggest future work could improve biological plausibility—perhaps through more advanced neuron models—without losing the tractability of thermodynamic interpretations. Can you provide a specific example of how this can be done?

*In the revised manuscript, we added an analysis of model specification using a model that captures the sparse and widespread activity of neural populations, with an activation function that goes beyond the logistic/tanh form. Please see our reply to “*Limited Discussion of Methodological and Modeling Limitations.*”*

We believe that this model itself provides a concrete example of how to capture important aspects of the biophysical behavior of real neurons. Other modeling frameworks that induce non-canonical activation functions include curved neural

networks, where deformed exponential families have been introduced. These energy-based models could be further extended to include asymmetric couplings, potentially with asynchronous updates. Accordingly, we added the following sentence at the end of the model limitations paragraph in the Discussion:

We believe that this model itself gives a specific example that can capture an important aspect of biophysical behavior of actual neurons. This energy-based models could be extended to include asymmetric couplings. Accordingly, we added the following sentence in the end of the model limitation paragraph in Discussion:

For example, the alternating-shrinking higher-order interaction model (Eq.~\ref{eq:alternating_shrinking_hoi_model}) could be extended to include asymmetric couplings, potentially with asynchronous updates in continuous-time limit.

Incomplete Use of Thermodynamic Measures: The second limitation concerns the scope of the metrics used. The study focuses on entropy flow as a proxy for time-asymmetry, but this alone does not fully capture the system's dynamical regime. For instance, high entropy flow could reflect different phenomena, such as proximity to criticality or chaos. The authors propose that complementary measures (like conditional entropy) would enrich the interpretive power of their analyses. Can you be more specific about this?

Thank you very much for this suggestion and for correctly noting the two possible interpretations of high entropy flow. To enrich the interpretation, as described earlier, we incorporated conditional entropy and reverse conditional entropy into the perturbation analysis (Fig.9B,C). These measures revealed subcritical behavior during steady activity: both decreased with increasing β , indicating that forward and reverse processes become more deterministic. This suggests that the system transitions from a disordered phase toward a ferromagnetic phase, rather than entering a quasi-chaotic regime as would be expected near a chaotic phase transition. Accordingly, we added the following sentence:

Both forward and reverse conditional entropies ($\sigma_{t^{\text{forward}}}$ and $\sigma_{t^{\text{backward}}}$) in Eq.~\ref{eq:ef_forward_reverse_decomp}) decreased with increasing β , indicating that both processes became more deterministic. This trend suggests that, as β increases, the system transitions from a disordered phase toward a ferromagnetic phase, rather than into a quasi-chaotic regime, as expected at a phase transition\cite{aguilera2021unifying}. Thus, these results indicate that the subsampled neural population during this period does not operate in a quasi-deterministic chaotic regime, but rather in a subcritical regime.

Since we have included the analysis of conditional entropy, we excluded the following sentences in Discussion.

Third, while we examined entropy flow in this study, assessing it alone provides an incomplete picture of the system. For example, increased entropy flow may occur when the system approaches criticality or a quasi-deterministic chaotic regime \cite{aguilera2023nonequilibrium}. Future studies could incorporate additional metrics, such as conditional entropy, for a more precise characterization of neural population states.

Again, we sincerely thank the reviewer for this valuable suggestion, which prompted us to provide a more precise interpretation of the dynamical regime of the observed neural population.

Temporal Scope of the Analysis: The third limitation involves the timescale of the analysis. The model is tested primarily on short timescales, suitable for stimulus-response dynamics, but it remains unclear how well it generalizes to longer-term processes, such as learning or memory consolidation. This suggests a need for future work on scaling the model to capture broader temporal dynamics. How can this be done in practice?

We agree that extending this analysis to longer-term processes, such as learning and memory consolidation, is an important direction, especially since entropy flow reflects the input–output causal responsiveness of neurons, a quantity that underlies Hebbian plasticity in neural systems and is expected to change during learning.

In practice, the current state-space framework can be applied to such data by assuming stationarity within sessions, treating time bins as trials, and allowing parameters to vary across sessions. This approach requires stable recordings from the same neurons across days and weeks, which is becoming increasingly feasible with both calcium imaging and large-scale electrophysiological recordings [1,2]. Under these conditions, the state-space method would provide a powerful means to trace learning trajectories. The approach becomes even more straightforward when more data are available (i.e., trials per time bin), since the stationarity assumption within sessions can then be satisfied through selective data adoption.

We agree that extending this analysis to longer-term processes, such as learning and memory consolidation, is an important direction, especially since entropy flow reflects the input–output causal responsiveness of neurons that may underlies Hebbian plasticity and is expected to change during learning. In practice, the current state-space framework can be applied to such data by assuming stationarity within sessions, treating time bins as trials, and allowing parameters to vary across sessions. This approach requires stable recordings from the same neurons across days and weeks, which is becoming increasingly feasible with both calcium imaging and large-scale

electrophysiological recordings [1,2]. Under these conditions, the state-space method would provide a powerful means to trace learning trajectories.

To highlight this point, we included the following clarification in the Discussion of the revised manuscript:

In addition, our framework could be extended to analyze longer-term processes such as learning by treating time bins as trials within sessions and allowing parameters to vary across sessions, under the assumption of stationarity within each session. This would enable tracing learning trajectories of couplings among individual neurons when stable longitudinal recordings are available, an increasingly feasible scenario with recent advances in calcium imaging and electrophysiology\cite{steinmetz2021neuropixels,van2025tracking}.

Accordingly, the following statement was omitted

Fourth, this analysis primarily focused on short timescales related to stimulus responses, and further research is needed to assess the model's ability to analyze longer-term dynamics, such as learning.

[1] Steinmetz, N. A., Aydin, C., Lebedeva, A., Okun, M., Pachitariu, M., Bauza, M., ... & Harris, T. D. (2021). Neuropixels 2.0: A miniaturized high-density probe for stable, long-term brain recordings. *Science*, 372(6539), eabf4588.

[2] van Beest, E. H., Bimbard, C., Fabre, J. M., Dodgson, S. W., Takács, F., Coen, P., ... & Carandini, M. (2025). Tracking neurons across days with high-density probes. *Nature Methods*, 22(4), 778-787.

3 Minor

• fig 2 B) and 5B) Estimated coupling parameters..this figures are not understandable, too saturated and it is not cclear the message you want to convey here.

We thank the reviewer for this helpful observation. We agree that the previous versions of Fig.2B and Fig.5B (now Fig.6B) were visually dense. Following your suggestion, we redesigned both panels to reduce saturation to show that time-varying second-order couplings can be reliably estimated.

Fig. 2B (simulation): For each selected neuron, we now show **five couplings** together with the ground-truth trajectory (black dashed line) and intervals around the estimate. This side-by-side view makes accuracy and dynamics directly readable.

Fig.6B (real data): For each reference neuron i (now $i=1,2,80$), we display at most 5 coupling trajectories. Candidates are first filtered by a pointwise interval test within an analysis window (bins 21–75): an interval must exclude zero at least once. Among the

surviving candidates, we show the smallest j labels to avoid cherry-picking very large effects and to keep the ordering deterministic.

- Fig 3. Estimation error and computational time. these results should be interpreted beyond the technique, they should illuminate a discussion the limitations and broader applications of this method.

The time and accuracy limitations may constrain the applicability of the method to large-scale neuroimaging signals such as fMRI or ECoG. To clarify this point, we added the following sentences in the new section discussing the broader applications of the method:

However, the state-space method still faces limits in computational time and scale, constraining its use for these large-scale signals. Future improvements through parallelization, optimized algorithms, and refined mean-field approaches could extend its applicability and enhance entropy flow estimation.

- Fig 5A. Is there an active condition? The time points should have a different color.
We made the histogram for active condition previously overlaid by the passive condition visible. We also removed confusing time points marked in red that indicated the timing for panel C.
- Is it possible to assign units to entropy flow?
At the introduction of entropy flow, we added:
Because we use the natural logarithm, we report entropy flow in units of nats.
- In the abstract "violating steady-state assumption." to "...violating the steady-state assumption."
Corrected
- In the abstract "...directed activity emerged in a fewer neurons..." "...directed activity emerged in fewer neurons..."
Corrected
- Introduction, line 69: " enabled to reveal details..." to "Thus, the method enabled us to reveal details..."
Corrected
- Results, Figure 2C caption: "...comparing the true coupling papameters..." to "...comparing the true coupling parameters..."
Corrected
- Results, line 248: Remove space before comma.
We could not find comma in this line.
- Discussion, sentence 519: "...statistical constructs and do not represent physiological synap-

tic connections." to "...statistical constructs that do not necessarily represent physiological synaptic connections."

Corrected

- "Fig. 5A Left" to "Fig. 5A (left)"

Corrected

Black: Reviewer's comments

Blue: Our responses

Reviewer #3:

However, it would further strengthen the manuscript if the authors could more explicitly delineate how future experimental manipulations (e.g., pharmacological perturbations, optogenetic modulation of inhibitory/excitatory balance) might serve to test the predictions derived from entropy flow analyses.

Following the two predictions on efficient information coding by neurons with higher entropy flow per spike already included in the Discussion, we added an additional prediction regarding neuromodulatory manipulations:

Moreover, targeted pharmacological or optogenetic manipulations of neuromodulatory systems are predicted to induce systematic changes in entropy flow by modulating coupling variability, thereby altering coding efficiency.

Referee Report, manuscript NCOMMS-25-20836

May 7, 2025

Title : *State-space kinetic Ising model reveals task-dependent entropy flow in sparsely active nonequilibrium neuronal dynamics*

Submitted for publication in Nature Communications

1 Manuscript overview

This manuscript introduces a state-space kinetic Ising model to capture nonstationary and nonequilibrium features in neuronal population activity. The authors propose a mean-field approximation to estimate time-varying entropy flow and apply the method to large-scale neural recordings from mouse visual cortex under active and passive behavioral states. The study is innovative in its methodological development and its application to understanding entropy dissipation dynamics linked to behavioral engagement.

While methodologically innovative and analytically rich, the manuscript currently falls short in conceptual framing and mechanistic insight. It proposes a technically interesting model, but lacks a fully developed hypothesis or critical discussion of generality and its limitations. Strengthening these aspects would significantly enhance its contribution to computational neuroscience and statistical physics of neural systems.

Overall, this article presents an interesting methodological approach; however, its applicability to a broader range of neuroscientific data appears limited. Furthermore, the reported results do not seem particularly compelling due to their high variability. Figures focusing on individual cases, such as Mouse ID: 574078, would be more appropriate in the supplementary material, while results pertaining to the broader population should be emphasized in the main text.

The current state of the manuscript presents important limitations in terms of hypothesis development, mechanistic interpretation, and transparency regarding model limitations. The work needs to address major issues before further consideration, for example the following.

2 Major

- **Limited Discussion on the Hypothesis Underpinning the Methodological Approach**

The manuscript provides minimal theoretical or empirical justification for why entropy flow (specifically as defined through the kinetic Ising model) should reflect meaningful cognitive or behavioral phenomena in neuronal populations.

There is no formal hypothesis on what the entropy flow dynamics are expected to reveal about the neural computation or behavior prior to analysis. As a result, the interpretation of increased entropy flow in high-performing mice remains post hoc and correlational.

I would suggest to clearly formulate a hypothesis about the expected relationship between entropy flow, spiking sparsity, and cognitive performance, possibly grounded in sparse coding or predictive coding frameworks.

- **Limited Discussion of Methodological and Modeling Limitations** The model's coupling parameters are interpreted as proxies for causal interactions, yet the manuscript ac-

knowledges that these do not map onto synaptic or effective connectivity. There is no attempt to systematically analyze or bound the interpretability of these couplings.

While the EM-based inference and mean-field approximations are interesting, the impact of model assumptions (e.g., binning, Gaussian process priors, independence of neurons' parameter dynamics) on the validity of entropy estimates is not rigorously evaluated.

The model is benchmarked only on simulation data that closely matches its own assumptions (e.g., Gaussian sampling of time-varying parameters). There is no analysis of robustness under model misspecification, e.g., when spiking statistics deviate from logistic assumptions or exhibit higher-order correlations.

I suggest to include a more explicit limitations subsection addressing identifiability, model misspecification, and the limits of using statistical constructs to infer neurobiological phenomena.

- **Limited Mechanistic Insight into Observed Neural Phenomena**

The observed association between increased entropy flow in high-firing neurons and behavioral performance is not mechanistically unpacked. Are these neurons more connected? Do they drive others? Are they receiving stronger input? The current model lacks the granularity to answer these questions.

Similarly, the interpretation of entropy flow as a proxy for “causal” information transfer or “organized structure” lacks specific neurophysiological anchoring. The term “structure” remains vague.

There is no model-based perturbation analysis to assess how altering coupling distributions or firing sparsity would predictably affect entropy flow. This weakens claims about the mechanistic role of sparse coding in modulating nonequilibrium dynamics.

I would suggest to integrate or at least discuss how this framework could be extended to, multiscale analyses, or mechanistic circuits. Can you comment on other types of data in neuroscience. EEG, MEG, fMRI, LFP's, MEA?

- **Better explain the limitations**

Physiological Interpretation of Couplings: The first limitation emphasizes that the model's estimated couplings, while useful for capturing directed dynamics, are statistical abstractions and not direct analogs of real synaptic connections. This disconnect is attributed to the simplified nonlinearity in the model (e.g., logistic or tanh functions), which does not fully capture the complex, biophysical behavior of actual neurons. The authors suggest future work could improve biological plausibility—perhaps through more advanced neuron models—without losing the tractability of thermodynamic interpretations. Can you provide a specific example of how this can be done?

Incomplete Use of Thermodynamic Measures: The second limitation concerns the scope of the metrics used. The study focuses on entropy flow as a proxy for time-asymmetry, but this alone does not fully capture the system's dynamical regime. For instance, high entropy flow could reflect different phenomena, such as proximity to criticality or chaos. The authors propose that complementary measures (like conditional entropy) would enrich the interpretive power of their analyses. Can you be more specific about this?

Temporal Scope of the Analysis: The third limitation involves the timescale of the analysis. The model is tested primarily on short timescales, suitable for stimulus-response dynamics, but it remains unclear how well it generalizes to longer-term processes, such as learning or memory consolidation. This suggests a need for future work on scaling the model to capture broader temporal dynamics. How can this be done in practice?

3 Minor

- fig 2 B) and 5B) Estimated coupling parameters..this figures are not understandable, too saturated and it is not cclear the message you want to convey here.
- Fig 3. Estimation error and computational time. these results should be interpreted beyond the technique, they should illuminate a discussion the limitations and broader applications of this method.
- Fig 5A. Is there an active condition? The time points should have a different color.
- Is it possible to assign units to entropy flow?
- In the abstract "violating steady-state assumption." to "...violating the steady-state assumption."
- In the abstract "...directed activity emerged in a fewer neurons..." "...directed activity emerged in fewer neurons..."
- Introduction, line 69: " enabled to reveal details..." to "Thus, the method enabled us to reveal details..."
- Results, Figure 2C caption: "...comparing the true coupling papameters..." to "...comparing the true coupling parameters..."
- Results, line 248: Remove space before comma.
- Discussion, sentence 519: "...statistical constructs and do not represent physiological synaptic connections." to "...statistical constructs that do not necessarily represent physiological synaptic connections."
- "Fig. 5A Left" to "Fig. 5A (left)"